# ADVERSARIAL TRAINING FOR DEFENSE AGAINST LABEL POISONING ATTACKS

**Melis Ilayda Bal**[1]***, Volkan Cevher**[2,3]**, Michael Muehlebach**[1]
[1]Max Planck Institute for Intelligent Systems, Tübingen, Germany
[2]LIONS, EPFL    [3]AGI Foundations, Amazon
`{mbal, michaelm}@tuebingen.mpg.de,`
`volkan.cevher@epfl.ch, volkcevh@amazon.de`

## ABSTRACT

As machine learning models grow in complexity and increasingly rely on publicly sourced data, such as the human-annotated labels used in training large language models, they become more vulnerable to label poisoning attacks. These attacks, in which adversaries subtly alter the labels within a training dataset, can severely degrade model performance, posing significant risks in critical applications. In this paper, we propose FLORAL, a novel adversarial training defense strategy based on support vector machines (SVMs) to counter these threats. Utilizing a bilevel optimization framework, we cast the training process as a non-zero-sum Stackelberg game between an *attacker*, who strategically poisons critical training labels, and the *model*, which seeks to recover from such attacks. Our approach accommodates various model architectures and employs a projected gradient descent algorithm with kernel SVMs for adversarial training. We provide a theoretical analysis of our algorithm's convergence properties and empirically evaluate FLORAL's effectiveness across diverse classification tasks. Compared to robust baselines and foundation models such as RoBERTa, FLORAL consistently achieves higher robust accuracy under increasing attacker budgets. These results underscore the potential of FLORAL to enhance the resilience of machine learning models against label poisoning threats, thereby ensuring robust classification in adversarial settings.

## 1 INTRODUCTION

The susceptibility of machine learning models to the integrity of their training data is a growing concern, particularly as these models become more complex and reliant on large volumes of publicly sourced data, such as the human-annotated labels used in training large language models (Kumar et al., 2020; Cheng et al., 2020; Wang et al., 2023). Any compromise in training data can severely undermine a model's performance and reliability (Dalvi et al., 2004; Szegedy et al., 2013), with potentially catastrophic consequences in high-stakes domains including fraud detection (Fiore et al., 2019), medical diagnosis (Finlayson et al., 2019), biological design (Bal et al., 2024b), and autonomous driving (Deng et al., 2020).

One of the most insidious forms of threat is the *data poisoning* (*causative*) attacks (Barreno et al., 2010), where adversaries subtly manipulate a subset of the training data, causing the model to learn erroneous input-output associations. These attacks can involve either feature or label perturbations. Unlike feature poisoning, which alters the input data itself, (*triggerless*) label poisoning is particularly challenging to detect because only the labels are modified, leaving the input data unchanged, as illustrated in Figure 2. Deep learning models are inherently vulnerable to random label noise (Zhang et al., 2017), and this susceptibility is magnified when the noise is adversarially crafted to be more damaging. Figure 1b illustrates this vulnerability: The RoBERTa model (Liu et al., 2019) fine-tuned for sentiment analysis suffers substantial performance degradation under label poisoning attacks (Zhu et al., 2022), with severity growing as the attacker's budget increases. In contrast, Figure 1c highlights FLORAL's effectiveness in mitigating these attacks. Here, the adversarially labelled dataset is generated by poisoning the labels of the most influential training points (see Appendix C.3 for details).

---

*Corresponding author. Code is available at `https://github.com/melisilaydabal/floral`.

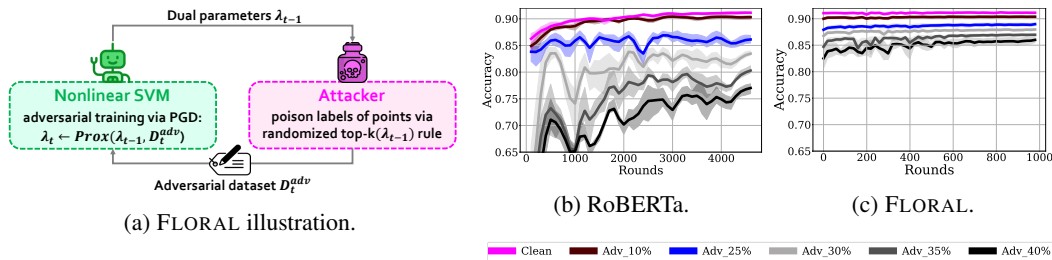

Figure 1: (a): The illustration of FLORAL defense, adversarial training under label poisoning attacks. (b): The test accuracy degradation of RoBERTa fine-tuned on the IMDB dataset with adversarial labels, showing its vulnerability to such attacks. (c): FLORAL effectively mitigates the impact of label poisoning in (b), achieving significantly higher robust accuracy.

A line of work has addressed label poisoning through designing triggerless attacks against SVMs (Biggio et al., 2012; Xiao et al., 2012; 2015), backdoor attacks in vision contexts (Chen et al., 2022; Jha et al., 2023) or combining label poisoning with adversarial attacks (Fowl et al., 2021; Geiping et al., 2021). Defense mechanisms typically focus on filtering (sanitization) techniques (Laishram & Phoha, 2016; Paudice et al., 2018), kernel correction (Biggio et al., 2011), intrinsic dimensionality-based sample weighting (Weerasinghe et al., 2021) or robust learning (Steinhardt et al., 2017). Adversarial training (AT) (Goodfellow et al., 2015; Madry et al., 2017) is a widely adopted empirical defense against data poisoning—particularly for feature perturbations—framing the interaction as a zero-sum game and training models on adversarially perturbed data (Huang et al., 2015; Kurakin et al., 2016). However, as shown in our experiments (Section 4.1), conventional AT does not adequately defend against label poisoning attacks, and its direct application to label poisoning remains largely unexplored.

In this paper, we address robust classification under label poisoning attacks and introduce FLORAL (**F**lipping **L**abels f**or** **A**dversarial **L**earning), an SVM-based adversarial training defense that can be seamlessly adapted to other model architectures. We formulate our defense strategy as a bilevel optimization problem (Robey et al., 2024), enabling a computationally *efficient* generation of optimal label attacks, and forming a non-zero-sum Stackelberg game between an *attacker* (or *adversary*), targeting critical training labels, and the *model*, recovering from such attacks. We propose a projected gradient descent algorithm tailored for kernel SVMs to solve the bilevel optimization problem. Our experiments on various classification tasks demonstrate that FLORAL improves robustness in the face of adversarially manipulated labels by effectively leveraging the inherent robustness of SVMs combined with the strengths of adversarial training, achieving enhanced model resilience against label poisoning while maintaining a balance with classification accuracy.

**Contributions.** Our main contributions are the following.

- We propose FLORAL, a support vector machine-based adversarial training strategy that defends against label poisoning attacks. To the best of our knowledge, this is the first work to introduce adversarial training as a defense specifically for *label poisoning attacks*. We consider kernel SVMs in our formulation, however, as we show in our experiments, the method can be easily integrated with other models such as neural networks.

- We utilize a bilevel optimization formulation for the robust learning problem, leading to a non-zero-sum Stackelberg game between an *attacker* who poisons the labels of influential training points and the *model* trying to recover from such attacks. We provide a projected gradient descent (PGD)–based algorithm to solve the game efficiently.

- We theoretically analyze the local asymptotic stability of our algorithm by proving that its iterative updates remain bounded and characterizing its convergence to the Stackelberg equilibrium.

- We empirically analyze FLORAL's effectiveness through experiments on various classification tasks against robust baselines as well as foundation models such as RoBERTa. Our results demonstrate that as the attacker's budget increases, FLORAL maintains higher robust accuracy compared to baselines trained on adversarial data.

- Finally, we show the generalizability of FLORAL against attacks from the literature, `alfa`, `alfa-tilt` (Xiao et al., 2015) and `LFA` (Paudice et al., 2018), which aim to maximize the difference in empirical risk between classifiers trained on tainted and untainted label sets.

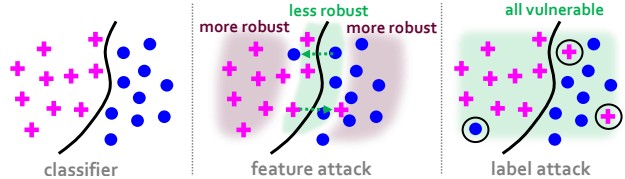

Figure 2: **Sensitivity of the decision boundary to label poisoning attacks.** The vulnerability of data points differs between feature perturbation and label poisoning attacks. Given a perfect classifier, points near the decision boundary are less robust to feature attacks (Zhang et al., 2021; Xu et al., 2023), leading to localized shifts in classification regions when the attack is performed. In contrast, the decision boundary has a broader sensitivity with respect to label poisoning attacks which can affect both near-boundary and distant points. By injecting incorrect labels, these attacks can create more widespread disruption and an overall degradation in classifier performance across the input space.

## 2 PROBLEM STATEMENT AND BACKGROUND

We tackle the problem of robust binary classification in the presence of label poisoning attacks (see Section 3 for an extension to multi-class classification). Given a training dataset $\mathcal{D} = \{(x_i, y_i) \in (\mathcal{X}, \mathcal{Y})\}_{i=1}^n$, where $\mathcal{X} \subseteq \mathbb{R}^d$ are the input features and $\mathcal{Y} = \{\pm 1\}$ are the binary labels (potentially involving adversarial labels), we consider a kernel SVM classifier $f_\lambda(x) := \text{sign}(\sum_j \lambda_j y_j k(x, x_j) + b)$, parametrized by $\lambda \in \mathbb{R}^n$ and bias $b \in \mathbb{R}$, which assigns a label to each data point and is derived from the following quadratic program (dual formulation) (Boser et al., 1992; Hearst et al., 1998):

$$D(f_\lambda; \mathcal{D}): \min_{\lambda \in \mathbb{R}^n} \quad \frac{1}{2}\lambda^{\mathrm{T}}Q\lambda - \mathbb{1}^{\mathrm{T}}\lambda \tag{1}$$

$$\text{subject to} \quad y^{\mathrm{T}}\lambda = 0, \quad 0 \le \lambda \le C, \tag{2}$$

where $Q \in \mathbb{R}^{n \times n}$ is a positive semi-definite matrix, with elements $Q_{ij} = y_i y_j K_{ij}$ and $\mathbb{1}$ is the $n$-dimensional vector of all ones. Here, $K$ is the Gram matrix with entries $K_{ij} = k(x_i, x_j), \forall i, j \in [n] := \{1, \ldots, n\}$, derived from a kernel function $k$. A common kernel choice is the radial basis function (RBF), given as $k(x_i, x_j) = \exp(-\gamma \|x_i - x_j\|^2)$, with width parameter $\gamma$. The parameter $C \ge 0$ is a regularization term, balancing the trade-off between maximizing the margin and minimizing classification errors. In this formulation, each dual variable $\lambda_i, i \in [n]$ corresponds to the Lagrange multiplier associated with the misclassification constraint for the training point $x_i$.

## 3 THE FLORAL APPROACH

In the context of label poisoning attacks, the attacker's objective is to maximize the model's test classification error by subtly altering the labels in the training dataset to an optimal adversarial configuration. Adversarial training (Goodfellow et al., 2015; Madry et al., 2017) can be extended to counter these attacks and minimize model sensitivity to disruptive labels by actively optimizing for robustness under worst-case scenarios. In this setting, the attacker generates the optimal label attack within a budget of $k$ flips to maximize the model's loss, while the model seeks parameters that minimize this worst-case loss. A straightforward, yet naive (Robey et al., 2024), way to implement this approach would be to use the following minimax formulation:

$$\min_{\lambda \in \mathbb{R}^n} \frac{1}{n} \sum_{i=1}^n \left\{ \max_{\substack{\sum_{i \in [n]} \mathbb{1}\{y_i \ne \tilde{y}_i\} = k \\ \tilde{y}_i \in \mathcal{Y}, i \in [n]}} \mathcal{L}\left(f_\lambda(x_i), \tilde{y}_i\right) \right\}, \tag{3}$$

where $\mathcal{L}$ denotes a loss function, which in the case of the kernel SVM is related to the hinge loss (Smola & Schölkopf, 1998), and $\tilde{y}$ represents the adversarial label set. This formulation is problematic for multiple reasons:

1. *Misaligned objectives*: The loss is only a surrogate for the test accuracy, which is the actual quantity of interest to both the learner and the attacker. However, from an optimization perspective, maximizing an upper bound (such as the hinge loss in SVMs) on the classification error as in (3) is not meaningful as such a bound does not represent the true objective of the attacker. Hence, a non-zero-sum formulation would allow for a more nuanced representation of the attacker's objectives (Yasodharan & Loiseau, 2019).

---

**Algorithm 1** FLORAL

---

1: **Input:** Initial kernel SVM model $f_{\lambda_0}$, training dataset $\mathcal{D}_0 = \{(x_i, y_i)\}_{i=1}^n, x_i \in \mathbb{R}^d, y_i \in \{\pm 1\}$, attacker budget $B \in \{0, \ldots, n\}$, parameter $k$, where $k \ll B$, learning rate $\eta > 0$.
2: **for** round $t = 1, \ldots, T$ **do**
3:    $\tilde{y}^t \leftarrow$ Solve (7-9) via randomized top-$k$ : randomly selecting $k$ points from top $B$ w.r.t. $\lambda_{t-1}$.
4:    $\mathcal{D}_t \leftarrow \{(x_i, \tilde{y}_i^t)\}_{i=1}^n$    */ Adversarial dataset with selected $k$ poisoned labels
5:    Compute gradient of the objective (4), $\nabla_\lambda D(f_{\lambda_{t-1}}; \mathcal{D}_t)$, based on $\lambda_{t-1}, \mathcal{D}_t$ as given in (10).
6:    Take a PGD step $\lambda_t \leftarrow \text{PROX}_{\mathcal{S}(\tilde{y}^t)}(\lambda_{t-1} - \eta \nabla_\lambda D(f_{\lambda_{t-1}}; \mathcal{D}_t))$, based on (11-12).    */ AT
7: **end for**
8: **return** $f_{\lambda_T}$

---

2. *Ineffective defense against critical points*: In the case of an SVM-based classifier, the minimax formulation would only safeguard against attacks targeting data points with the largest hinge loss, i.e., those farthest from the decision boundary. These attacks are easily distinguishable (Xiao et al., 2012) as, e.g., soft margin SVMs are shown to be robust to outliers (Smola & Schölkopf, 1998). In contrast, attacks targeting the critical points that define the decision boundary (*support vectors*) would be more effective in degrading the classifier's performance.

3. *Combinatorial explosion*: Even if a bilevel formulation is employed where the attacker minimizes the margin, the problem remains computationally challenging. Ordering data points by their margin and then searching for the best adversarial label set within a budget constraint results in a vast combinatorial space.

As a result of these, we formulate our adversarial training routine as a non-zero-sum Stackelberg game (Von Stackelberg, 2010; Conitzer & Sandholm, 2006) and propose FLORAL defense using the bilevel optimization formulation (Bard, 2013):

$$D(f_\lambda; \mathcal{D}): \min_{\lambda \in \mathbb{R}^n} \quad \frac{1}{2}\lambda^\mathrm{T}\tilde{Q}\lambda - \mathbb{1}^\mathrm{T}\lambda \qquad (4)$$

$$\text{where } \tilde{y}(\lambda) \in \arg \max_{y' \in \mathcal{Y}^n, u \in \{0,1\}^n} \lambda^\mathrm{T} u \qquad (7)$$

$$\text{subject to} \quad \tilde{y}(\lambda)^\mathrm{T}\lambda = 0 \quad (5)$$

$$\text{subject to} \quad y_i' = y_i(1 - 2u_i), \forall i \in [n] \quad (8)$$

$$0 \leq \lambda \leq C \quad (6)$$

$$\sum_{i \in [n]} \mathbf{1}\{y_i \neq y_i'\} = k. \quad (9)$$

In the outer (model's) problem, defined by (4-6), the SVM classifier is derived under an adversarial label set. The key difference from the formulation in Section 2 is that the elements of $\tilde{Q}$ are defined as $\tilde{Q}_{ij} = \tilde{y}_i\tilde{y}_j K_{ij}$. Meanwhile, the inner (attacker's) problem, given by (7-9) identifies the top-$k$ most *influential* data points affecting the model's decision boundary. The intuition behind this approach is similar to identifying the most responsible training points for the model's prediction as in (Koh & Liang, 2017). However, rather than relying on influence functions (Hampel, 1974), the attacker leverages the dual variables $\lambda$, which provides *direct* access to such influential points. These points correspond to the support vectors, and the higher the value of a dual variable, the more critical that data point is in determining the model's decision boundary.

We address the bilevel optimization problem in (4-9) as a non-zero-sum Stackelberg game (Von Stackelberg, 2010) between the learning *model*, and the *attacker* acting as the leader and follower, respectively, as shown in Figure 1a. The game begins with an initial kernel SVM model $f_{\lambda_0}$ and a training dataset $\mathcal{D}_0$, and proceeds iteratively. In each round $t$, the model shares its dual parameters with the attacker, who then generates an adversarially labelled dataset $\mathcal{D}_t$ using a *randomized top-k* rule. That is, the attacker identifies the top-$B$ data points based on their $\lambda_{t-1}$ values, (constrained by the budget $B$) and flips the labels of $k$ randomly chosen points among them. We incorporate randomization to account for the attacker's budget and to reduce the risk of settling in local optima. Adversarial training is performed via a projected gradient descent step using $\lambda_{t-1}$ and $\mathcal{D}_t$, after which the updated parameters, $\lambda_t$, are shared with the attacker. This iterative interplay between the attacker and defender model forms a soft-margin kernel SVM robust to adversarial label poisoning. Our overall approach is detailed in Algorithm 1.

**FLORAL's effectiveness.** FLORAL iteratively exposes the model to learn adversarial configurations of the decision boundary. Hence, when the training data is clean, the training process *proactively* adjusts the model to be less sensitive to the influence of individual poisoned labels. In cases where poisoned labels are already present in the initial training data, FLORAL effectively neutralizes their impact by *implicitly sanitizing* the corrupted labels. This behavior is evaluated empirically and detailed in Section 4.1 and Appendix D.

**The attacker's capability.** The attacker solves (7-9) with respect to the shared model parameters $\lambda$, generating label attacks by targeting the most influential support vectors. This white-box attack (Wu et al., 2023) assumes that the attacker can access model parameters. To reflect practical constraints, we limit the attacker's budget to at most $B$ label poisons per round, from which $k$ points are randomly selected. While this scenario may still seem to give the attacker significant power, notably, $(i)$ relying on secrecy for security is generally considered poor practice (Biggio et al., 2013), and $(ii)$ our method is designed to defend against the strongest possible attacker. Even in black-box attack scenarios, where the attacker lacks parameter access, FLORAL remains effective for generating transferable attacks (Zheng et al., 2023). In such cases, the attacker could fit a kernel SVM on the available data and apply a similar selection rule to craft adversarial labels.

**Gradient of the objective (4).** In each round, the adversarial training PGD step requires computing the gradient $\nabla_\lambda D(f_\lambda; \mathcal{D})$ of the objective (4) based on $\lambda_{t-1}$ and $\mathcal{D}_t$, which is defined as

$$\nabla_\lambda D(f_{\lambda_{t-1}}; \mathcal{D}_t) = \tilde{Q}\lambda_{t-1} - \mathbb{1}, \tag{10}$$

where $\tilde{Q}$ is the matrix with entries $\tilde{Q}_{ij} = \tilde{y}_i^t \tilde{y}_j^t K_{ij}, \forall i, j \in [n]$, detailed in Appendix B.

**Projection.** The feasible set $\mathcal{S}$ changes in each round $t$ depending on the adversarial label set $\tilde{y}^t$ (see (6)). We introduce the variable $z_t := \lambda_{t-1} - \eta\nabla_\lambda D(f_{\lambda_{t-1}}; \mathcal{D}_t)$ and define the projection operator $\text{PROX}_{\mathcal{S}(\tilde{y}^t)}(z_t) : \mathbb{R}^n \to \mathbb{R}^n$ as follows:

$$\text{PROX}_{\mathcal{S}(\tilde{y}^t)}(z_t) : \lambda_t \in \arg\min_{\lambda \in \mathbb{R}^n} \quad \frac{1}{2}\|\lambda - z_t\|^2 \tag{11}$$

$$\text{subject to} \quad \tilde{y}^{t^{\text{T}}}\lambda = 0, \quad 0 \le \lambda \le C. \tag{12}$$

However, solving this quadratic program for large-scale instances is computationally challenging unless the specific problem structure is exploited. Therefore, we provide a scalable and efficient implementation of Algorithm 1 that relies on a fixed point iteration strategy as detailed in Section 3.2.

**A form of geometry-aware AT.** FLORAL aligns with geometry-aware AT principles (Zhang et al., 2021). Support vectors with large Lagrange multipliers ($\lambda$) play a critical role in defining the decision boundary (Hearst et al., 1998). In FLORAL, the attacker strategically identifies these points using a randomized top-$k$ rule. This method inherently integrates the geometric proximity to the decision boundary into the label attack, targeting points that significantly impact the hinge loss.

**Robust multi-class classification.** We extend our algorithm to multi-class classification tasks, as detailed in Algorithm 3 in Appendix F. The primary modification involves adopting a one-vs-all approach and considering multiple attackers, each corresponding to a different class.

## 3.1 STABILITY ANALYSIS

We theoretically analyze the stability of FLORAL (Algorithm 1) by $(i)$ demonstrating that its iterative updates are bounded and $(ii)$ characterizing its convergence to the Stackelberg equilibrium. For simplicity of notation, let us define the update rule at round $t$ as $\lambda_t := \text{PROX}_{\mathcal{S}(y_t)}(z_t) = \text{PROX}_{\mathcal{S}(y_t)}(\lambda_{t-1} - \eta\nabla_\lambda f(\lambda_{t-1}, y_t))$, where PROX is defined in (11-12). We use the operator LFLIP : $\mathcal{X} \times \mathcal{Y} \to \mathcal{Y}$ to define label poisoning attack formulated in (7-9).

**Lemma 1.** *Let $(\hat{\lambda}, \hat{y}(\hat{\lambda}))$ denote a Stackelberg equilibrium, i.e., $\hat{y}(\hat{\lambda}) := \text{LFLIP}(\hat{\lambda})$ and $\hat{\lambda} := \text{PROX}_{\mathcal{S}(\hat{y}(\hat{\lambda}))}(\hat{z}) = \text{PROX}_{\mathcal{S}(\hat{y}(\hat{\lambda}))}(\hat{\lambda} - \eta\nabla_\lambda f(\hat{\lambda}, \hat{y}(\hat{\lambda})))$ and $\{\lambda_t\}_{t=0}^T$ be the sequence of iterates generated by FLORAL (Algorithm 1). The following bound holds for the iterates:*

$$\|\lambda_t - \hat{\lambda}\|_\infty \le \|z_t - \hat{z}\|_\infty + \kappa_y\|y_t - \hat{y}(\hat{\lambda})\|_\infty \tag{13}$$

*where $\kappa_y$ is a constant defined by the PROX operator and index set corresponding to $\lambda_t \in (0, C)$, as detailed in Appendix A.1, and $\|\cdot\|_\infty$ denotes the infinity norm.*

*Proof.* See Appendix A.1 for the proof.

**Lemma 2.** *Let $(\hat{\lambda}, \hat{y}(\hat{\lambda}))$ denote the Stackelberg equilibrium as before. The following bound holds for the non-projected iterates $\{z_t\}_{t=0}^T$ of* FLORAL *(Algorithm 1):*

$$\|z_t - \hat{z}\|_\infty \leq \kappa_\lambda \|\lambda_{t-1} - \hat{\lambda}\|_\infty + \kappa'_y \|y_t - \hat{y}(\hat{\lambda})\|_\infty \tag{14}$$

*where $\kappa_\lambda$ and $\kappa'_y$ are kernel dependent constants that are below 1 for small enough $\eta$.*

*Proof.* See Appendix A.2 for the proof.

**Theorem 3.1** ($\varepsilon$-local asymptotic stability)**.** *The Stackelberg equilibrium $(\hat{\lambda}, \hat{y}(\hat{\lambda}))$ defined as before, is $\varepsilon$-locally asymptotically stable for the Stackelberg game solved via Algorithm 1 for a small enough step size $\eta$. This implies that for every $\varepsilon > 0$, there exists $\delta > 0$ such that*

$$\|\lambda_0 - \hat{\lambda}\|_\infty < \delta \Rightarrow \|\lambda_t - \hat{\lambda}\|_\infty < \varepsilon, \forall t > 0 \text{ and } \lambda_t \to \hat{\lambda}. \tag{15}$$

*Proof (sketch).* The proof relies on characterizing the distance between the update $\lambda_t$ at round $t$ and the equilibrium $\hat{\lambda}$ using Lemma 1 and Lemma 2, then leveraging the fact that the label flipping operator (LFLIP) formulated in (7-9) returns the same adversarial label set when $\lambda_t$ is within an $\varepsilon$ distance from the equilibrium. The complete proof is given in Appendix A.3, with the global convergence result discussed in Appendix A.4.

## 3.2 LARGE-SCALE IMPLEMENTATION

We scale our algorithm for large problem instances by approximating the projection operation (step 6 in Algorithm 1) via a fixed-point iteration method, as outlined in Algorithm 2. The key idea leverages the optimal $\lambda^\star$ expression from Appendix A.1 and involves an iterative splitting of variables based on non-projected $\lambda$ values within the range $[0, C]$. In each iteration, the variable $\mu$ is updated using the expression in Appendix A.1 until convergence to a specified error $\epsilon$ is achieved.

---

**Algorithm 2** PROJECTIONVIAFIXEDPOINTITERATION

1: **Input:** Non-projected $\lambda_0$, adversarial label set $\tilde{y} = \{\tilde{y}_i\}_{i=1}^n, y_i \in \{\pm 1\}$, parameters $\{C, \epsilon\} > 0$.
2: Initialize $\mu_0 = 0$.
3: **for** round $t = 1, \ldots, T_{\text{proj}}$ **do**
4:     $\lambda_t = \text{CLIP}_{[0,C]}(\lambda_0 - \mu_{t-1}\tilde{y})$            */ Clip to satisfy constraint in (12)
5:     **if** $\lambda_t \tilde{y} = 0$ **then**
6:         **return** $\lambda_t$
7:     **end if**
8:     $\mathcal{I}_C, \mathcal{I}_z \leftarrow$ indices of $\lambda_t \geq C, \lambda_t \in (0, C)$            */ Variable splitting
9:     $\eta \leftarrow \max(|\mathcal{I}_z|, 1)$                                  */ To avoid empty $\mathcal{I}_z$ case
10:    $\mu_t \leftarrow \frac{\eta - |\mathcal{I}_z|}{\eta}\mu_{t-1} + \frac{1}{\eta}(\sum_{i \in \mathcal{I}_C} C\tilde{y}_i + \sum_{i \in \mathcal{I}_z} \lambda_t^i \tilde{y}_i)$
11:    **if** $|\mu_t - \mu_{t-1}| \leq \epsilon$ **then**
12:        **return** $\text{CLIP}_{[0,C]}(\lambda_0 - \mu_t \tilde{y})$
13:    **end if**
14: **end for**

---

## 3.3 RELATED WORK

**Label poisoning.** Biggio et al. (2012) first analyzed label poisoning attacks, showing that flipping a small number of training labels severely degrades SVM performance. Xiao et al. (2012) later formalized optimal label flip attacks under budget constraints as a bilevel optimization problem, which then expanded to transferable attacks on black-box models (Zhao et al., 2017), considering arbitrary attacker objectives. Beyond SVMs, recent works have explored label poisoning in backdoor attack scenarios, where adversaries inject triggers or alter triggerless data with poisoned labels in multi-label settings (Jha et al., 2023; Chen et al., 2022). In contrast, our approach focuses on triggerless poisoning attacks.

Defenses against these attacks include heuristic-based kernel correction (Biggio et al., 2011), which uses expectation for $Q$ in (4), though assuming independent label flipping with equal probability–a condition not guaranteed in practice. Other defenses such as clustering-based filtering (Laishram & Phoha, 2016; Tavallali et al., 2022), data complexity analysis (Chan et al., 2018), re-labeling (Paudice

et al., 2018) and label smoothing (Rosenfeld et al., 2020) offer straightforward solutions, however, they do not scale well to high-dimensional or large datasets. Sample weighting based on local intrinsic dimensionality (LID) (Weerasinghe et al., 2021; Ma et al., 2018) shows promise, but relies on accurate and computationally expensive LID estimation. Our approach, however, avoids strong assumptions about the data distribution or the attacker, preserves feasibility, and scales effectively to large-scale problem instances as demonstrated in Section 4. Additionally, while learning under noisy labels (Frénay & Verleysen, 2013; Natarajan et al., 2013; Hallaji et al., 2023; Zhang et al., 2024) may seem relevant, our work focuses specifically on *adversarial* label noise (Biggio et al., 2011), where the adversary *intentionally* crafts the most damaging label perturbations.

**Adversarial training (AT).**  Adversarial examples, introduced by Szegedy et al. (2013), revealed how small perturbations cause misclassification in deep neural networks (DNNs). Building on this, AT (Goodfellow et al., 2015) emerged as a prominent defense, training models on both original and adversarially perturbed data. Defenses have utilized adversarial examples generated by methods such as the Fast Gradient Sign Method (FGSM) (Goodfellow et al., 2015), PGD (Madry et al., 2017), Carlini & Wagner attack (Carlini & Wagner, 2017), among others (Chen et al., 2017; Moosavi-Dezfooli et al., 2015). For SVMs, Zhou et al. (2012) formulated convex AT for linear SVMs, later extended to kernel SVMs by Wu et al. (2021) via doubly stochastic gradients under feature perturbations. Despite this progress, AT for label poisoning remains underexplored. FLORAL fills this gap, by leveraging AT specifically for label poisoning scenarios, using PGD to train models on poisoned datasets rather than generating adversarial examples.

In parallel, game-theoretical approaches have modeled adversarial interactions as simultaneous games (Bal et al., 2024a), where classifiers and adversaries select strategies independently (Dalvi et al., 2004), or as Stackelberg games with a leader-follower dynamic (Brückner & Scheffer, 2011; Zhou et al., 2019; Chivukula et al., 2020). AT has further linked these concepts, particularly in simultaneous zero-sum games (Hsieh et al., 2019; Pinot et al., 2020; Pal & Vidal, 2020) to non-zero-sum formulations (Robey et al., 2024). We adopt a sequential setup, using the Stackelberg framework where the leader commits to a strategy and the follower responds accordingly.

## 4 EXPERIMENTS

In this section, we showcase the effectiveness of FLORAL across various robust classification tasks, utilizing the following datasets:

- **Moon** (Pedregosa et al., 2011): We employed a synthetic benchmark dataset, $\mathcal{D} = \{(x_i, y_i)\}_{i=1}^{2000}$ where $x_i \in \mathbb{R}^2$ and $y_i \in \{\pm 1\}$. Adversarial versions are generated by flipping the labels of points farther from the decision boundary of a linear classifier trained on the clean dataset, using label poisoning levels (%) of $\{5, 10, 15, 20, 25\}$. The details on the adversarial datasets are given in Appendix C.1.
- **IMDB** (Maas et al., 2011): A benchmark review sentiment analysis dataset with $\mathcal{D} = \{(x_i, y_i)\}_{i=1}^{50000}$ where $x_i \in \mathbb{R}^{768}$ and $y_i \in \{\pm 1\}$. For SVM training, we extracted 768-dimensional embeddings from the fine-tuned RoBERTa (Liu et al., 2019). We created adversarial datasets by fine-tuning the RoBERTa-base model on the clean dataset to identify influential training points based on the gradient with respect to the inputs, then flipping their labels at poisoning levels (%) of $\{10, 25, 30, 35, 40\}$.
- **MNIST** (Deng, 2012): In Appendix E.3, we provide the additional experiments with the MNIST dataset in detail.

**Experimental setup.**  For all SVM-based methods, we used RBF kernel, exploring various values of $C$ and $\gamma$. We conducted five replications with different train/test splits, including the corresponding adversarial datasets for each dataset. In all FLORAL experiments, we constrain the attacker's capability with a limited budget. That is, the attacker identifies the most influential *candidate* points, with $B = 2k$, from the training set and randomly selects $k \in \{1, 2, 5, 10, 25\}$ to poison, where $k$ represents the % of points relative to the training set size. Detailed experimental configurations are provided in Appendix C (see Table 3).

**Baselines.**  We benchmark FLORAL against the following baselines:

1. (Vanilla) SVM with an RBF kernel, which serves as a basic benchmark (Hearst et al., 1998).
2. LN-SVM (Biggio et al., 2011) applies a heuristic-based kernel matrix correction.

Table 1: Test accuracies of methods trained on the Moon dataset, averaged over five runs. Highlighted values indicate the best performance in the "**Best**" (peak accuracy during training) and "Last" (final accuracy after training) columns. This notation is consistently applied in the subsequent tables. FLORAL outperforms baselines in most of the settings, providing a particularly robust defense in highly adversarial scenarios. See Appendix E.1 (Table 4) for the results of other settings.

| Setting | | FLORAL | | SVM | | NN | | NN-PGD | | LN-SVM | | Curie | | LS-SVM | | K-LID | |
|---|---|---|---|---|---|---|---|---|---|---|---|---|---|---|---|---|---|
| | | Best | Last | Best | Last | Best | Last | Best | Last | Best | Last | Best | Last | Best | Last | Best | Last |
| Clean | $C=10, \gamma=1$ | **0.968** | 0.966 | 0.968 | 0.968 | 0.960 | 0.960 | 0.966 | 0.964 | 0.940 | 0.940 | 0.941 | 0.941 | 0.881 | 0.881 | 0.966 | 0.966 |
| $D^{adv}=5\%$ | $C=10, \gamma=1$ | **0.966** | 0.966 | 0.965 | 0.957 | 0.926 | 0.926 | 0.964 | 0.937 | 0.940 | 0.940 | 0.903 | 0.903 | 0.881 | 0.881 | 0.964 | 0.964 |
| $D^{adv}=10\%$ | $C=10, \gamma=1$ | 0.924 | 0.907 | 0.912 | 0.900 | 0.859 | 0.855 | **0.927** | 0.853 | 0.869 | 0.868 | 0.907 | 0.907 | 0.894 | 0.894 | 0.908 | 0.907 |
| $D^{adv}=15\%$ | $C=10, \gamma=1$ | **0.924** | 0.917 | 0.892 | 0.823 | 0.826 | 0.826 | 0.871 | 0.871 | 0.893 | 0.829 | 0.906 | 0.858 | 0.892 | 0.823 | 0.883 | 0.826 |
| $D^{adv}=20\%$ | $C=10, \gamma=1$ | **0.875** | 0.865 | 0.840 | 0.771 | 0.788 | 0.787 | 0.854 | 0.853 | 0.839 | 0.758 | 0.859 | 0.763 | 0.840 | 0.771 | 0.830 | 0.755 |
| $D^{adv}=25\%$ | $C=10, \gamma=1$ | **0.801** | 0.768 | 0.753 | 0.717 | 0.693 | 0.647 | 0.740 | 0.655 | 0.754 | 0.693 | 0.779 | 0.697 | 0.766 | 0.721 | 0.747 | 0.690 |

Table 2: Test accuracies of methods trained on the IMDB dataset, averaged over five replications. FLORAL demonstrates superior robustness compared to baselines, particularly in more adversarial scenarios. See also Figures 1b-1c.

| Setting | FLORAL | | RoBERTa | | SVM | | LN-SVM | | Curie | | LS-SVM | | K-LID | |
|---|---|---|---|---|---|---|---|---|---|---|---|---|---|---|
| | Best | Last | Best | Last | Best | Last | Best | Last | Best | Last | Best | Last | Best | Last |
| Clean | 0.9113 | 0.9113 | **0.9119** | 0.9110 | 0.9113 | 0.9113 | 0.9113 | 0.9113 | 0.9116 | 0.9113 | 0.9108 | 0.9108 | 0.9116 | 0.9115 |
| $D^{adv}=10\%$ | 0.9039 | 0.9039 | **0.9048** | 0.9031 | 0.9039 | 0.9039 | 0.9029 | 0.9028 | 0.9039 | 0.9039 | 0.9010 | 0.9010 | 0.9039 | 0.9039 |
| $D^{adv}=25\%$ | **0.8896** | 0.8896 | 0.8827 | 0.8612 | 0.8887 | 0.8886 | 0.8860 | 0.8860 | 0.8889 | 0.8888 | 0.8771 | 0.8769 | 0.8885 | 0.8883 |
| $D^{adv}=30\%$ | **0.8801** | 0.8801 | 0.8675 | 0.8357 | 0.8792 | 0.8792 | 0.8771 | 0.8771 | 0.8797 | 0.8797 | 0.8325 | 0.8324 | 0.8795 | 0.8795 |
| $D^{adv}=35\%$ | **0.8713** | 0.8713 | 0.8270 | 0.8053 | 0.8660 | 0.8660 | 0.8646 | 0.8646 | 0.8695 | 0.8695 | 0.7667 | 0.7667 | 0.8700 | 0.8700 |
| $D^{adv}=40\%$ | **0.8636** | 0.8636 | 0.7792 | 0.7717 | 0.8574 | 0.8584 | 0.8515 | 0.8515 | 0.8589 | 0.8589 | 0.7060 | 0.7060 | 0.8594 | 0.8594 |

3. Curie (Laishram & Phoha, 2016), utilizes the DBSCAN clustering (Ester et al., 1996) to identify and filter-out poisoned data points.
4. LS-SVM (Paudice et al., 2018) applies label sanitization based on k-NN (Cover & Hart, 1967).
5. K-LID (Weerasinghe et al., 2021), a weighted SVM based on kernel local intrinsic dimensionality.
6. NN: A DNN trained using the SGD optimizer with momentum, serving as a non-linear baseline.
7. NN-PGD: A DNN trained with PGD-AT (Madry et al., 2017), to evaluate a robust model designed to withstand feature perturbation attacks under label poisoning.
8. RoBERTa (Liu et al., 2019), used in experiments with IMDB dataset to assess a fine-tuned transformer-based language model's robustness under label poisoning.

Appendix G includes further comparisons with a least squares classifier using randomized smoothing (Rosenfeld et al., 2020) and a filtering-out defense based on regularized synthetic reduced nearest neighbor (Tavallali et al., 2022) on the Moon and MNIST datasets.

**Performance metrics.** We assess our method using two key metrics: robust and clean accuracy, tracked over a test set with *clean labels* during training. Unlike feature perturbation studies, where robust accuracy is gauged on adversarially perturbed test examples (Yang et al., 2020), in our study, robust accuracy reflects model performance tested on clean labels using adversarially labelled training data, thereby indicating the models' resilience and generalization capabilities under poisoning. Conversely, clean accuracy measures the performance of models trained and tested on clean-labelled data, offering a benchmark for comparison under both adversarial and non-adversarial conditions. We additionally report hinge loss on the clean-labelled test data (see Appendix E.2) in experiments with the IMDB dataset.

## 4.1 EXPERIMENT RESULTS

In this section, we report the performance of FLORAL against the baseline methods on the Moon dataset, followed by the results of its integration with RoBERTa on the IMDB dataset.

**Moon.** As reported in Table 1 and Figure 3, FLORAL achieves higher robust accuracy across almost all settings compared to baseline methods. Notably, in scenarios with more severe poisoning levels, FLORAL significantly outperforms all baselines, which experience a marked drop in their accuracy. We report results under various kernel hyperparameters in Appendix E.1 (see Table 4 and

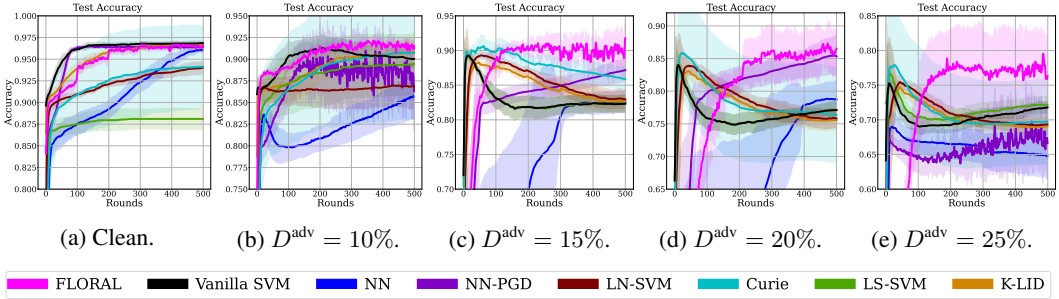

Figure 3: Test accuracy of methods on the Moon dataset under varying label poisoning levels. For SVM models, $C = 10$, $\gamma = 1$ are used. See Appendix E (Figure 7) for results with other settings. As the label poisoning level increases, the accuracy of methods generally declines, however, FLORAL maintains higher robust accuracy across all adversarial settings, without compromising clean accuracy.

Figure 7). We additionally visualize the decision boundaries of trained methods on the test dataset in Figure 4, which shows that FLORAL produces a smoother decision boundary compared to the baselines, promoting generalization (see Figure 18 in Appendix E for the complete results).

When the initial training data is clean, FLORAL provides a *proactive defense* by introducing adversarial labels during training, thereby effectively reducing the model's sensitivity to potential label attacks. Notably, FLORAL matches performance on par with vanilla SVM on clean data, demonstrating that its robust framework maintains high accuracy without compromising clean accuracy. In scenarios with already poisoned training data, FLORAL achieves robustness through two key mechanisms: $(i)$ implicitly sanitizing corrupted labels, while $(ii)$ introducing additional adversarial labels to further reduce model sensitivity to attacks. We analyze this sanitization effect in detail in Appendix D and show that FLORAL sanitizes $25 - 35\%$ portion of the initial poisoned training labels. Furthermore, FLORAL operates on *dynamically evolving* adversarial datasets during training, unlike baselines that are trained on fixed adversarially labelled datasets. This dynamic strategy introduces new adversarial labels in each round, further testing and enhancing the model's robustness. These capabilities position FLORAL as a superior defense over baselines, particularly in maintaining robust accuracy under more challenging adversarial scenarios.

FLORAL offers several distinct advantages over existing baselines, e.g., unlike LN-SVM, FLORAL does not rely on the strong assumption that training labels are independently flipped with equal probability. Compared to Curie, FLORAL avoids a filter-out system that risks discarding data with valuable feature representations. Further, in terms of scalability, Curie's dependence on distance metrics makes it vulnerable to the curse of dimensionality, diminishing its clustering performance in high-dimensional and complex datasets. To ensure a fair comparison, we calibrated the noise, confidence level, and threshold value parameters of LN-SVM, Curie, and LS-SVM baselines, aligning them with the poisoning level in each dataset. This ensures that the baselines are at their strongest configurations, highlighting the robustness and scalability of FLORAL.

**IMDB.** We integrate FLORAL as a robust classifier head for RoBERTa. The test performance on the IMDB dataset, shown in Table 2 and Figures 1b-1c, reveals that FLORAL significantly improves robustness, outperforming fine-tuned RoBERTa along with other baselines. Our approach also converges faster to lower loss values, in more adversarial scenarios (see Table 5 and Figure 8 in Appendix E.2).

In Appendix E.2, we analyze the changes in influential training points—those that most affect model predictions—when applying FLORAL to RoBERTa-extracted embeddings (see Figures 9-10). The results reveal some overlap in the identified points, under clean train data scenario. However, as the training data becomes more adversarial, FLORAL identifies different critical points, which effectively shape the decision boundary and contribute to improved robust accuracy.

**Adaptability.** As shown with the IMDB experiments, FLORAL integrates seamlessly with other model architectures (e.g., RoBERTa, NNs) by utilizing the last-layer embeddings for training. We additionally demonstrate this in Appendix I, FLORAL integrated with an NN learns more robust representations and achieves higher robust accuracy on the Moon and MNIST datasets.

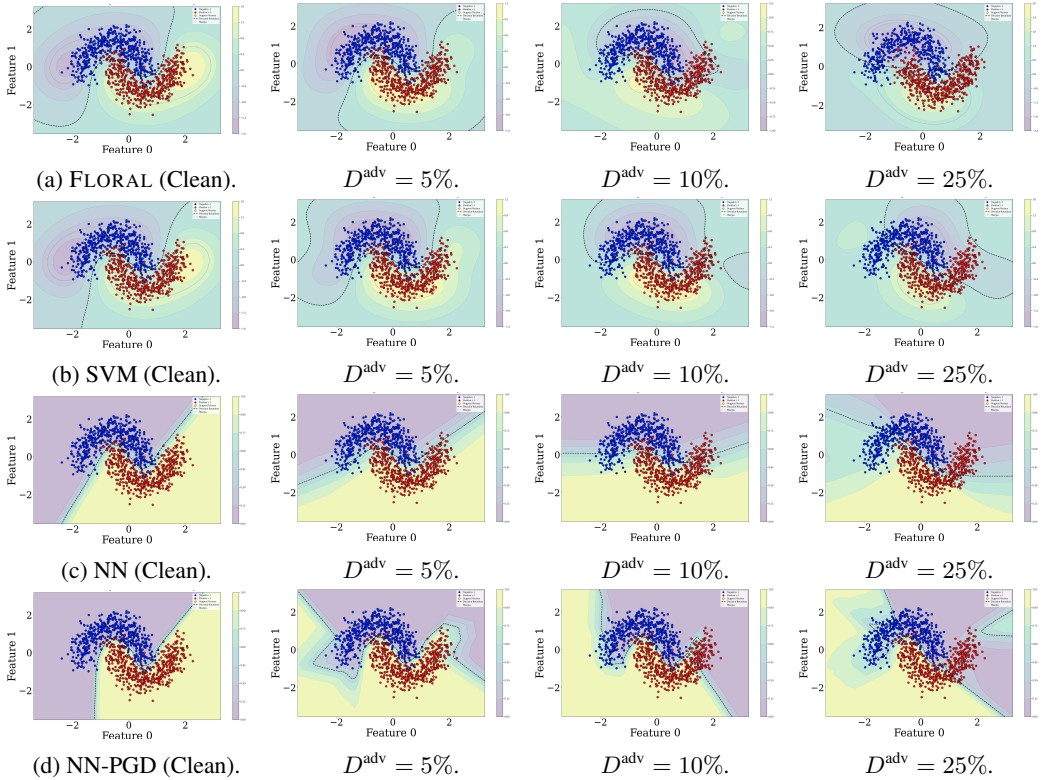

Figure 4: The decision boundaries on the Moon test dataset under varying label poisoning levels. SVM models use an RBF kernel with $C = 10$ and $\gamma = 0.5$. FLORAL generates a smooth decision boundary compared to baseline methods, which show drastic changes due to adversarial training label manipulations. For the complete results with other baselines, see Appendix E (Figure 18).

**Sensitivity analysis.** We further examined the sensitivity of our approach to the attacker's budget, and the results are detailed in Appendix E.4 with Figure 12.

**Generalizability.** We demonstrate the effectiveness of FLORAL under different attacks: `alfa`, `alfa-tilt` (Xiao et al., 2015) and `LFA` (Paudice et al., 2018) in Appendix H. Our experiments on the Moon and MNIST (Deng, 2012) datasets again confirmed that FLORAL can also defend and achieve higher robust accuracy in the presence of other types of label attacks.

**Limitations.** Defense strategies may not be universally effective against all label poisoning attacks due to their non-adaptive nature (Papernot et al., 2016). Our defense strategy relies on a white-box attack, where the attacker can access the model. While we also show the performance of our approach under various label attacks from the literature, its efficacy may vary under different attack scenarios.

## 5 CONCLUSION

In this paper, we address the vulnerability of machine learning models to label poisoning attacks and propose FLORAL, an adversarial training defense strategy based on kernel SVMs. We formulate the problem using bilevel optimization and frame the adversarial interaction between the learning model and the attacker as a non-zero-sum Stackelberg game. To compute the game equilibrium that solves the optimization problem, we introduce a projected gradient descent-based algorithm and analyze its local stability and convergence properties. Our approach demonstrates superior empirical robustness across various classification tasks compared to robust baseline methods.

Future research includes exploring SVM-based transfer attacks or integrating our approach to robust fine-tuning of foundation models for supervised downstream tasks. Additionally, a detailed analysis of how FLORAL alters the most influential training points for model predictions, e.g. when integrated with foundation models such as RoBERTa could provide interesting insights.

ACKNOWLEDGEMENTS

This research was supported by the Max Planck & Amazon Science Hub. We also thank the German Research Foundation for the support and Zhiyu He for the helpful comments on the manuscript. The work was conducted during Volkan Cevher's time at Amazon.

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

# Appendix

## Table of Contents

# A    THEORETICAL ANALYSIS PROOFS

In this section, we present the proofs for the local asymptotic stability analysis of FLORAL (Algorithm 1). We begin by proving Lemma 1 in Section A.1, which establishes that the distance of the updates of Algorithm 1 from the equilibrium of the game is bounded. In Section A.2, we prove Lemma 2, demonstrating that the distance of the non-projected updates from the equilibrium of the game is also bounded. Lastly, in Section A.3, we provide the proof of Theorem 3.1, which shows the local asymptotic stability of our algorithm, with a derivation of a global convergence result presented in Section A.4.

## A.1    PROOF OF LEMMA 1

Our objective is to prove that the distance of the iterates of Algorithm 1 from the Stackelberg equilibrium $(\hat{\lambda}, \hat{y}(\hat{\lambda}))$, specifically $\lambda_t - \hat{\lambda}$, is bounded. We begin by recalling the update rule at round $t$, $\lambda_t := \text{PROX}_{\mathcal{S}(y_t)}(z_t) = \text{PROX}_{\mathcal{S}(y_t)}(\lambda_{t-1} - \eta \nabla_\lambda f(\lambda_{t-1}, y_t))$, where $y_t = \hat{y}(\lambda_{t-1})$, $\mathcal{S}(y_t)$ is the feasible region defined by constraints (12), using the labels at round $t$. The operator PROX is defined below.

**Definition 1** (PROX operator). *The operator $\text{PROX}_{\mathcal{S}(y_t)}(z_t) : \mathbb{R}^n \to \mathbb{R}^n$ denotes the projection of $z_t \in \mathbb{R}^n$ onto the convex set $\mathcal{S}(y_t)$ at round $t$ of Algorithm 1. PROX minimizes the Euclidean distance and is defined by the following optimization problem:*

$$\text{PROX}_{\mathcal{S}(y_t)}(z_t) : \lambda_t \in \arg\min_{\lambda \in \mathbb{R}^n} \quad \frac{1}{2}\|\lambda - z_t\|^2 \tag{16}$$

$$\text{subject to} \quad y_t^{\mathrm{T}}\lambda = 0 \tag{17}$$

$$0 \le \lambda \le C \tag{18}$$

*Equivalently, $\text{PROX}_{\mathcal{S}(y_t)}$ solves the following optimization problem:*

$$\min_{\substack{\lambda \in \mathbb{R}^n \\ 0 \le \lambda \le C}} \quad \sup_{\mu \in \mathbb{R}} \quad \frac{1}{2}\|\lambda - z_t\|^2 + \mu y_t^{\mathrm{T}}\lambda. \tag{19}$$

**Lemma 3** (Bounded iterates). *The sequence $\{\lambda_t\}$ generated by the iterative update rule $\lambda_t := \text{PROX}_{\mathcal{S}(y_t)}(z_t) = \text{PROX}_{\mathcal{S}(y_t)}(\lambda_{t-1} - \eta \nabla_\lambda f(\lambda_{t-1}, y_t))$ is bounded, i.e., $\|\lambda_t\|_\infty \le C, \forall t \ge 0$.*

*Proof.* This follows immediately from the definition of $\mathcal{S}(y_t)$. □

In the following, our aim is to quantify the sensitivity of (19) with respect to its arguments $y_t$ and $z_t$. Let $\lambda^\star$ denote the optimal solution to the projection operation. We can express this solution through the following steps. First, we simplify the expression by omitting the index $t$ in (19). Then, we exploit the fact that the objective function is convex-concave with convex constraints, which allows us to interchange the order of the min and the sup. This yields

$$\min_{\substack{\lambda \in \mathbb{R}^n \\ 0 \le \lambda \le C}} \quad \sup_{\mu \in \mathbb{R}} \quad \frac{1}{2}\|\lambda - z\|^2 + \mu y^{\mathrm{T}}\lambda$$

$$= \sup_{\mu \in \mathbb{R}} \quad \min_{\substack{\lambda \in \mathbb{R}^n \\ 0 \le \lambda \le C}} \quad \frac{1}{2}\|\lambda - z + \mu y\|^2 - \frac{1}{2}\mu^2\|y\|^2.$$

At this stage, the optimization problem over $\lambda$ reduces to the minimization of a quadratic function over box constraints. We can therefore express $\lambda^\star$ based on the optimal choice $\mu^\star$ for $\mu$ as follows:

$$\lambda_i^\star = \begin{cases} 0, & \text{if } z_i - \mu^\star(z,y)y_i \le 0 \\ z_i - \mu^\star(z,y)y_i, & \text{if } 0 < z_i - \mu^\star(z,y)y_i < C \\ C, & \text{if } z_i - \mu^\star(z,y)y_i \ge C \end{cases} \quad \text{and choose } \mu^\star(z,y) \text{ such that } y^{\mathrm{T}}\lambda^\star = 0,$$

$\forall i \in \{1, \dots, n\}$. We use the notation $\mu^\star(z,y)$ to highlight the dependency of the multiplier $\mu^\star$ on the variable $z$ and the label $y$.

We introduce the $\text{CLIP}_{[0,C]}(\cdot)$ operator which clips the value of the given input to the interval $[0, C]$. This operator yields the following compact expression for $\lambda^\star$:

$$\lambda^\star = \text{CLIP}_{[0,C]}(z - \mu^\star(z,y)y), \text{ where } \mu^\star(z,y) \text{ is chosen such that } y^{\mathrm{T}}\lambda^\star = 0. \tag{20}$$

By substituting the previous expression for $\lambda^\star$ into the equality constraint, we obtain

$$y^{\mathrm{T}}\mathrm{CLIP}_{[0,C]}(z - \mu^\star(z,y)y) = 0, \qquad (21)$$

which provides an equation that implicitly defines $\mu^\star(z,y)$. We further simplify (21) by indexing the components of $z - \mu^\star(z,y)y$ with respect to their values, which yields

$$
\begin{aligned}
0 &= \sum_{i \in \mathcal{I}_C} Cy_i + \sum_{i \in \mathcal{I}_z} y_i(z_i - \mu^\star(z,y)y_i) \\
&= \sum_{i \in \mathcal{I}_C} Cy_i + \sum_{i \in \mathcal{I}_z} z_iy_i - \sum_{i \in \mathcal{I}_z} \mu^\star(z,y)y_i^2 \\
&= \sum_{i \in \mathcal{I}_C} Cy_i + \sum_{i \in \mathcal{I}_z} z_iy_i - \sum_{i \in \mathcal{I}_z} \mu^\star(z,y). \qquad \text{(from } y_i^2 = 1)
\end{aligned}
$$

where $\mathcal{I}_z := \{i \mid \lambda_i = z_i - \mu^\star(z,y)y_i \in (0,C)\}$ with cardinality $\mid \mathcal{I}_z \mid$ and $\mathcal{I}_C := \{i \mid \lambda_i = z_i - \mu^\star(z,y)y_i \geq C\}$. We further solve for $\mu^\star$, which yields

$$\mu^\star(z,y) = \frac{1}{\mid \mathcal{I}_z \mid} \left( \sum_{i \in \mathcal{I}_C} Cy_i + \sum_{i \in \mathcal{I}_z} z_iy_i \right).$$

This equation implicitly defines $\mu^\star(z,y)$, which represents the basis for the fixed point iteration introduced in Algorithm 2.

This equation will also be the basis for computing sensitivities, i.e. quantifying how $\lambda^\star$ and $\mu^\star$ change when altering $z$ or $\lambda$. We first compute $\frac{\partial \lambda^\star}{\partial z}$. For a data point $i$, the following can be stated:

$$
\frac{\partial \lambda_i^\star}{\partial z} =
\begin{cases}
e_i^{\mathrm{T}} - \frac{\partial \mu^\star(z,y)}{\partial z}y_i, & \text{if } z_i - \mu^\star(z,y)y_i \in (0,C) \\
0, & \text{else,}
\end{cases}
\qquad (22)
$$

where $e_i$ is the $i$th standard basis vector. Differentiating the constraint (17) yields

$$
\begin{aligned}
0 &= \frac{\partial(y^{\mathrm{T}}\lambda^\star)}{\partial z} = \sum_{i=1}^{n} \frac{\partial \lambda_i^\star}{\partial z}y_i \\
&= \sum_{i \in \mathcal{I}_z} \left( e_i^{\mathrm{T}}y_i - \frac{\partial \mu^\star(z,y)}{\partial z}y_i^2 \right).
\end{aligned}
$$

Substituting $y_i^2 = 1$ into the previous equation yields

$$\frac{\partial \mu^\star(z,y)}{\partial z} = \frac{\sum\limits_{i \in \mathcal{I}_z} e_i^{\mathrm{T}}y_i}{\mid \mathcal{I}_z \mid}, \qquad (23)$$

where $\mathcal{I}_z$ with cardinality $\mid \mathcal{I}_z \mid$ is defined previously. From (22) and (23), we have

$$
\frac{\partial \lambda_i^\star}{\partial z} =
\begin{cases}
e_i^{\mathrm{T}} - \frac{\sum\limits_{j \in \mathcal{I}_z} e_j^{\mathrm{T}}y_j}{|\mathcal{I}_z|}y_i, & \text{if } i \in \mathcal{I}_z \\
0, & \text{if } i \notin \mathcal{I}_z.
\end{cases}
$$

Therefore, we conclude that

$$\left\| \frac{\partial \lambda_i^\star}{\partial z} \right\|_\infty \leq 1, \forall i \in [n], \qquad (24)$$

where we have exploited the fact that $y_i \in \{\pm 1\}$.

We further note that in the situation $\mathcal{I}_z = \emptyset$, $\lambda^\star \in \{0, C\}$, a change in $z$ or $y$ will not affect $\lambda^\star$ unless $z = \mu^\star y$ or $z = C + \mu^\star y$. As a result, we have for $\mathcal{I}_z = \emptyset$, $\frac{\partial \lambda^\star}{\partial z} = \frac{\partial \lambda^\star}{\partial y} = 0$ (a.e.).

We now compute $\frac{\partial \lambda^\star}{\partial y}$. For a data point $i$, the following holds:

$$
\frac{\partial \lambda_i^\star}{\partial y} =
\begin{cases}
-\frac{\partial \mu^\star(z,y)}{\partial y}y_i - e_i^{\mathrm{T}}\mu^\star(z,y), & \text{if } i \in \mathcal{I}_z \\
0, & \text{if } i \notin \mathcal{I}_z.
\end{cases}
\qquad (25)
$$

Differentiating the constraint (17) with respect to $y$ yields

$$
\begin{aligned}
0 = \frac{\partial(y^{\mathrm{T}}\lambda^{\star})}{\partial y} &= \lambda^{\star^{\mathrm{T}}} + \sum_{i=1}^{n} \frac{\partial \lambda_i^{\star}}{\partial y} y_i \\
&= \lambda^{\star^{\mathrm{T}}} + \sum_{i \in \mathcal{I}_z} \left( -\frac{\partial \mu^{\star}(z,y)}{\partial y} |y_i|^2 - y_i \mu^{\star}(z,y) e_i^{\mathrm{T}} \right).
\end{aligned}
$$

It follows from $|y_i|^2 = 1$ that

$$
\frac{\partial \mu^{\star}(z,y)}{\partial y} = \frac{\lambda^{\star^{\mathrm{T}}} - \mu^{\star}(z,y) \sum\limits_{i \in \mathcal{I}_z} y_i e_i^{\mathrm{T}}}{|\mathcal{I}_z|}. \tag{26}
$$

From (25) and (26), we obtain the following.

$$
\frac{\partial \lambda_i^{\star}}{\partial y} = \begin{cases} -\dfrac{y_i \lambda^{\star^{\mathrm{T}}}}{|\mathcal{I}_z|} + \mu^{\star}(z,y) \left( \dfrac{y_i \sum\limits_{j \in \mathcal{I}_z} e_j^{\mathrm{T}} y_j}{|\mathcal{I}_z|} - e_i^{\mathrm{T}} \right), & \text{if } i \in \mathcal{I}_z \\ 0, & \text{if } i \notin \mathcal{I}_z. \end{cases}
$$

As a result, we conclude using Lemma 3 that the following bound holds $\forall i \in [n]$

$$
\left\| \frac{\partial \lambda_i^{\star}}{\partial y} \right\|_{\infty} \leq \frac{\|\lambda\|_{\infty}}{|\mathcal{I}_z|} + |\mu^{\star}(z,y)| \leq \underbrace{\frac{C}{|\mathcal{I}_z|} + |\mu^{\star}|}_{\kappa_y}, \tag{27}
$$

where $\kappa_y$ is a constant that only depends on $C$ and the features of the dataset.

From (24) and (27), we conclude that

$$
\begin{aligned}
\|\lambda_t - \hat{\lambda}\|_{\infty} &= \|\lambda^{\star}(z_t, y_t) - \lambda^{\star}(z_t, \hat{y}(\hat{\lambda})) + \lambda^{\star}(z_t, \hat{y}(\hat{\lambda})) - \lambda^{\star}(\hat{z}, \hat{y}(\hat{\lambda}))\|_{\infty} \\
&\leq \kappa_y \|y_t - \hat{y}(\hat{\lambda})\|_{\infty} + \|z_t - \hat{z}\|_{\infty}.
\end{aligned}
$$

$\square$

## A.2 Proof of Lemma 2

Our objective is to prove that the distance of the non-projected updates of Algorithm 1 from the Stackelberg equilibrium $(\hat{\lambda}, \hat{y}(\hat{\lambda}))$, specifically $z_t - \hat{z}$, is bounded.

We begin by recalling the update rule at round $t$, $\lambda_t := \mathrm{PROX}_{\mathcal{S}(y_t)}(z_t) = \mathrm{PROX}_{\mathcal{S}(y_t)}(\lambda_{t-1} - \eta \nabla_{\lambda} f(\lambda_{t-1}, y_t))$, where $y_t = \hat{y}(\lambda_{t-1})$, $\mathcal{S}(y_t)$ is the feasible set defined by constraints (12), using the labels at round $t$. We further recall the Stackelberg equilibrium $(\hat{\lambda}, \hat{y}(\hat{\lambda}))$, i.e.,

$$
\hat{\lambda} := \mathrm{PROX}_{\mathcal{S}(\hat{y}(\hat{\lambda}))}(\hat{z}) = \mathrm{PROX}_{\mathcal{S}(\hat{y}(\hat{\lambda}))}(\hat{\lambda} - \eta \nabla_{\lambda} f(\hat{\lambda}, \hat{y}(\hat{\lambda})))
$$

$$
\hat{y}(\hat{\lambda}) := \mathrm{LFLIP}(\hat{\lambda}),
$$

where the operator $\mathrm{LFLIP} : \mathcal{X} \times \mathcal{Y} \to \mathcal{Y}$ defines the label poisoning attack formulated in (7-9). We conclude the following:

$$
\begin{aligned}
z_t &= \lambda_{t-1} - \eta \nabla_{\lambda} f(\lambda_{t-1}, y_t) \\
\hat{z} &= \hat{\lambda} - \eta \nabla_{\lambda} f(\hat{\lambda}, \hat{y}(\hat{\lambda})) \\
z_t - \hat{z} &= \lambda_{t-1} - \hat{\lambda} - \eta \left( \nabla_{\lambda} f(\lambda_{t-1}, y_t) - \nabla_{\lambda} f(\hat{\lambda}, \hat{y}(\hat{\lambda})) \right).
\end{aligned}
$$

We apply the mean value theorem for functions with multiple variables to the previous expression which allows us to rewrite $z_t - \hat{z}$ as

$$
\begin{aligned}
&= \lambda_{t-1} - \hat{\lambda} - \eta \left( \nabla_{\lambda} f(\lambda_{t-1}, y_t) - \nabla_{\lambda} f(\hat{\lambda}, y_t) + \nabla_{\lambda} f(\hat{\lambda}, y_t) - \nabla_{\lambda} f(\hat{\lambda}, \hat{y}(\hat{\lambda})) \right) \\
&= \lambda_{t-1} - \hat{\lambda} - \eta \left( \nabla_{\lambda}^2 f(\xi_{\lambda}, y_t)(\lambda_{t-1} - \hat{\lambda}) + \nabla_{\lambda y}^2 f(\hat{\lambda}, \xi_y)(y_t - \hat{y}(\hat{\lambda})) \right),
\end{aligned}
$$

where $\xi_{\lambda} \in (\hat{\lambda}, \lambda_{t-1})$ and $\xi_y \in (\hat{y}(\hat{\lambda}), y_t)$. The last equation can be restated as:

$$
z_t - \hat{z} = (I - \eta \nabla_{\lambda}^2 f(\xi_{\lambda}, y_t))(\lambda_{t-1} - \hat{\lambda}) - \eta \nabla_{\lambda y}^2 f(\hat{\lambda}, \xi_y)(y_t - \hat{y}(\hat{\lambda})), \tag{28}
$$

where $I$ denotes the identity matrix. We have defined the gradient of the objective in (10) as $\nabla_\lambda f(\lambda, y) = \tilde{Q}\lambda - \mathbb{1}$, where $\tilde{Q}$ is the matrix with entries $\tilde{Q}_{ij} = y_i y_j K_{ij}, \forall i, j \in [n]$, using the simplified notation. We express the second-order partial derivatives as:

$$\nabla_\lambda^2 f(\lambda; y) = K \odot yy^{\mathrm{T}}, \tag{29}$$

$$\nabla_{\lambda y}^2 f(\lambda; y) = K \odot y\lambda^{\mathrm{T}} + I \odot (K(\lambda \odot y)\mathbb{1}^{\mathrm{T}}), \tag{30}$$

where $K$ is the Gram matrix, $I$ is the $n \times n$ identity matrix, $\mathbb{1}$ is the all-one vector and $\odot$ denotes the Hadamard product. From (28), (29) and (30), we obtain

$$\begin{aligned}
z_t - \hat{z} = {} & (I - \eta\left((K \odot y_t y_t^{\mathrm{T}})\right)(\lambda_{t-1} - \hat{\lambda}) \\
& - \eta\left(K \odot \xi_y \hat{\lambda}^{\mathrm{T}} + I \odot (K(\hat{\lambda} \odot \xi_y)\mathbb{1}^{\mathrm{T}})\right)(y_t - \hat{y}(\hat{\lambda})).
\end{aligned}$$

We take the infinity norm and conclude:

$$\begin{aligned}
\|z_t - \hat{z}\|_\infty = {} & \|(I - \eta\left(K \odot y_t y_t^{\mathrm{T}})\right)(\lambda_{t-1} - \hat{\lambda}) \\
& - \eta\left(K \odot \xi_y \hat{\lambda}^{\mathrm{T}} + I \odot (K(\hat{\lambda} \odot \xi_y)\mathbb{1}^{\mathrm{T}})\right)(y_t - \hat{y}(\hat{\lambda}))\|_\infty \\
\leq {} & \|(I - \eta\left(K \odot y_t y_t^{\mathrm{T}})\right)(\lambda_{t-1} - \hat{\lambda})\|_\infty \\
& + \| - \eta\left(K \odot \xi_y \hat{\lambda}^{\mathrm{T}} + I \odot (K(\hat{\lambda} \odot \xi_y)\mathbb{1}^{\mathrm{T}})\right)(y_t - \hat{y}(\hat{\lambda}))\|_\infty \quad \text{(triangle inequality)} \\
= {} & \|(I - \eta\left(K \odot y_t y_t^{\mathrm{T}})\right)(\lambda_{t-1} - \hat{\lambda})\|_\infty \\
& + \|\eta\left(K \odot \xi_y \hat{\lambda}^{\mathrm{T}} + I \odot (K(\hat{\lambda} \odot \xi_y)\mathbb{1}^{\mathrm{T}})\right)(y_t - \hat{y}(\hat{\lambda}))\|_\infty \quad \text{(homogeneity)} \\
\leq {} & \underbrace{\|(I - \eta\left(K \odot y_t y_t^{\mathrm{T}})\right)\|_\infty}_{\kappa_\lambda} \|\lambda_{t-1} - \hat{\lambda}\|_\infty \\
& + \underbrace{\|\eta\left(K \odot \xi_y \hat{\lambda}^{\mathrm{T}} + I \odot (K(\hat{\lambda} \odot \xi_y)\mathbb{1}^{\mathrm{T}})\right)\|_\infty}_{\kappa_y'} \|y_t - \hat{y}(\hat{\lambda})\|_\infty. \quad \text{(homogeneity)}
\end{aligned}$$

This implies that

$$\|z_t - \hat{z}\|_\infty \leq \kappa_\lambda \|\lambda_{t-1} - \hat{\lambda}\|_\infty + \kappa_y' \|y_t - \hat{y}(\hat{\lambda})\|_\infty.$$

$\square$

We note that $\kappa_\lambda \leq 1$ if the learning rate $\eta$ is chosen small enough.

### A.3 Proof of Theorem 3.1

Let $(\hat{\lambda}, \hat{y}(\hat{\lambda}))$ denote the Stackelberg equilibrium, i.e.,

$$\hat{\lambda} := \text{PROX}_{\mathcal{S}(\hat{y}(\hat{\lambda}))}(\hat{z}) = \text{PROX}_{\mathcal{S}(\hat{y}(\hat{\lambda}))}(\hat{\lambda} - \eta\nabla_\lambda f(\hat{\lambda}, \hat{y}(\hat{\lambda})))$$

$$\hat{y}(\hat{\lambda}) := \text{LFLIP}(\hat{\lambda}),$$

where the operator $\text{LFLIP} : \mathcal{X} \times \mathcal{Y} \to \mathcal{Y}$ defines the label poisoning attack formulated in (7-9). We further assume that the LFLIP operator returns a unique set of adversarial labels at the Stackelberg equilibrium $(\hat{\lambda}, \hat{y}(\hat{\lambda}))$, which implies that there are no ties with respect to $\hat{\lambda}$ values. As a result, there exists a small enough constant $\delta' > 0$ such that for any $\lambda_0$ with $\|\lambda_0 - \hat{\lambda}\|_\infty < \delta'$, the corresponding $\hat{y}(\lambda_0)$ satisfies $\hat{y}(\lambda_0) = \hat{y}(\hat{\lambda})$. (Indeed, as long as $\delta'$ is small enough, such that the top-k entries between $\hat{\lambda}$ and $\lambda_0$ agree, $\hat{y}(\lambda_0) = \hat{y}(\hat{\lambda})$ will be satisfied.)

By combining Lemma 1 and Lemma 2 we conclude

$$\|\lambda_1 - \hat{\lambda}\|_\infty \leq \kappa_y \|\hat{y}(\lambda_0) - \hat{y}(\hat{\lambda})\|_\infty + \|z_1 - \hat{z}\|_\infty \leq \|z_1 - \hat{z}\|_\infty \leq \kappa_\lambda \|\lambda_0 - \hat{\lambda}\|_\infty < \kappa_\lambda \delta',$$

where we used the fact that $\hat{y}(\lambda_0) = \hat{y}(\hat{\lambda})$. The learning rate $\eta$ is chosen small enough, such that $\kappa_\lambda < 1$ and therefore $\|\lambda_1 - \hat{\lambda}\|_\infty < \kappa_\lambda \delta' < \delta'$. We therefore conclude by induction on $t$ that $\|\lambda_t - \hat{\lambda}\|_\infty < \kappa_\lambda^t \delta'$ for all $t > 0$. This readily implies $\lambda_t \to \hat{\lambda}$. Moreover, choosing $\delta = \min\{\epsilon, \delta'\}$ concludes $\|\lambda_t - \hat{\lambda}\|_\infty < \epsilon$ and concludes the proof. $\square$

### A.4 GLOBAL CONVERGENCE RESULT

The previous section provides the proof of Theorem 3.1, which provides a local stability and convergence result. Under additional assumptions on the constants $\kappa_y$ and $\kappa'_y$ that capture the sensitivity of the iterates $\lambda_t$ with respect to changes in the labels, one can derive a global convergence result, as summarized by the following proposition:

**Proposition 1.** *Let $(\hat{\lambda}, \hat{y}(\hat{\lambda}))$ denote the Stackelberg equilibrium as before and let $\delta' = (\hat{\lambda}_{\{k\}} - \hat{\lambda}_{\{k+1\}})/2 > 0$, where $\hat{\lambda}_{\{1\}}$ denotes the largest entry of $\hat{\lambda}$, $\hat{\lambda}_{\{2\}}$ the second larges entry of $\hat{\lambda}$, etc. Provided that*

$$\frac{2(\kappa_y + \kappa'_y)k}{1 - \kappa_\lambda} < \delta'$$

*holds and that the step-size $\eta$ is chosen to be small enough, the iterates $\{\lambda_t\}$ of* FLORAL *are guaranteed to converge to $\hat{\lambda}$ from any initial condition $\lambda_0$.*

*Proof.* As a result of Lemma 1 and Lemma 2 we conclude that

$$\|\lambda_t - \hat{\lambda}\|_\infty \leq \kappa_y \|\hat{y}(\lambda_{t-1}) - \hat{y}(\hat{\lambda})\|_\infty + \|z_{t-1} - \hat{z}\|_\infty$$
$$\leq (\kappa_y + \kappa'_y)\|\hat{y}(\lambda_{t-1}) - \hat{y}(\hat{\lambda})\|_\infty + \kappa_\lambda \|\lambda_{t-1} - \lambda_0\|_\infty.$$

We further take advantage of the fact that $\|\hat{y}(\lambda) - \hat{y}(\hat{\lambda})\|_\infty \leq 2k$ for any $\lambda$ (at most $k$ labels are flipped), which implies

$$\|\lambda_t - \hat{\lambda}\|_\infty \leq \kappa_\lambda \|\lambda_{t-1} - \lambda_0\|_\infty + 2(\kappa_y + \kappa'_y)k.$$

The previous inequality is satisfied for all $t$, and can be used to conclude that

$$\|\lambda_t - \hat{\lambda}\|_\infty \leq \kappa_\lambda^t \|\lambda_0 - \hat{\lambda}\|_\infty + \frac{2(\kappa_y + \kappa'_y)k}{1 - \kappa_\lambda} \tag{31}$$

holds for all $t$ (this can be verified by an induction argument). As a result, there exists an integer $t' > 0$ such that $\|\lambda_t - \hat{\lambda}\|_\infty < \delta'$ for all $t > t'$. This implies, due to the choice of $\delta'$, that $\hat{y}(\lambda_t) = \hat{y}(\hat{\lambda})$ for all $t > t'$. We therefore conclude that for all $t > t' + 1$

$$\|\lambda_t - \hat{\lambda}\|_\infty \leq \kappa_\lambda \|\lambda_{t-1} - \hat{\lambda}\|_\infty.$$

This readily implies $\lambda_t \to \hat{\lambda}$, due to the fact that $\kappa_\lambda < 1$, and implies the desired result. $\square$

## B THE GRADIENT OF THE OBJECTIVE (4)

We begin by recalling the kernel SVM dual formulation (Boser et al., 1992; Hearst et al., 1998):

$$D(f_\lambda; \mathcal{D}) : \min_{\lambda \in \mathbb{R}^n} \quad \frac{1}{2}\sum_{i=1}^n \sum_{j=1}^n \lambda_i \lambda_j y_i y_j K_{ij} - \sum_{i=1}^n \lambda_i$$

$$\text{subject to} \quad \sum_{i=1}^n \lambda_i y_i = 0$$

$$0 \leq \lambda_i \leq C, \forall i \in [n],$$

where $K$ represents the Gram matrix with entries $K_{ij} = k(x_i, x_j), \forall i, j \in [n] := \{1, \ldots, n\}$, derived from a kernel function $k$. We consider the $p^{\text{th}}$ data point and apply differentiation of a double summation to the objective, which yields

$$\frac{\partial (D(f_\lambda; \mathcal{D}))}{\partial \lambda_p} = \frac{1}{2}\left(\sum_{i=1}^n \lambda_i y_i y_p K_{ip} + \sum_{j=1}^n \lambda_j y_p y_j K_{pj}\right) - 1$$

$$= y_p \sum_{i=1}^n \lambda_i y_i K_{ip} - 1. \qquad \text{(from the symmetry of the kernel function)}$$

In compact form, we obtain the following.

$$\nabla_\lambda D(f_\lambda; \mathcal{D}) = Q\lambda - \mathbb{1},$$

where $Q$ is the matrix with entries $Q_{ij} = y_i y_j K_{ij}, \forall i, j \in [n]$ and $\mathbb{1}$ is the vector of all ones.

## C EXPERIMENT DETAILS

For our experiments, we set the hyperparameter values as given in Table 3. We provide the experiment details as follows.

- We initialize the model $f_{\lambda_0}$ with parameters set to 0. In FLORAL, however, the attacker uses a randomized top-$k$ rule to identify the $B$ most influential support vectors based on the $\lambda$ values. Due to the 0 initialization of $\lambda$, a warm-up period is required, which we set to 1 round for all SVM-related methods.
- To train kernel SVM classifiers for all SVM-related methods other than FLORAL, we use our PGD-based Algorithm 1 with a *dummy* attack, that is, we eliminate the adversarial dataset generation step and employ vanilla PGD training.
- For large datasets such as IMDB, we implement projection via fixed point iteration as given in Algorithm 2 in Section 3.2 instead of constructing a quadratic program as defined in (11-12).

Table 3: Hyperparameter values.

| Symbol | Hyperparameter | Value |
|---|---|---|
| $n$ | The size of the training dataset | Moon: 500, IMDB: 20000 |
| $T$ | The number of training rounds | Moon: 500, IMDB: 1000 |
| $T_{\text{proj}}$ | The number of projection via fixed point iteration rounds | 1000 |
| $B$ | The attacker budget | Moon: $\{10, 20, 50, 100, 250\}$, IMDB: $\{500, 2500, 5000, 12500\}$ |
| $k$ | The number of labels to poison | Moon: $\{5, 10, 25, 50, 125\}$, IMDB $\{250, 1250, 2500, 6250\}$ |
| $C$ | Regularization parameter for soft-margin SVM | Moon: $\{10, 100\}$, IMDB: 10 |
| $\gamma$ | RBF kernel parameter | Moon: $\{0.5, 1, 10\}$, IMDB: 0.005 |
| $\epsilon$ | Error rate for projection via fixed point iteration | $1e - 21$ |
| $\eta$ | Learning rate | Optimized over the set $\{0.0001, 0.0003, 0.0005, 0.0007\}$, for RoBERTa: $2e - 05$ |
| | Learning rate scheduler | Moon: a decay rate of 0.1 at every $\{100, 200\}$ rounds (optimized), for RoBERTa: linear scheduler |
| | The model architecture for NN and NN-PGD | Fully connected MLP with 2 hidden layers with 32 units each |
| | Batch size | 32 |
| | NN-PGD perturbation amount | 8/255 |
| | NN-PGD step size | 2/255 |
| | SGD optimizer momentum value | 0.9 |

### C.1 DATASETS

- Moon is a benchmark dataset for binary classification tasks, generated directly using the `scikit-learn` library (Pedregosa et al., 2011). It contains two-dimensional input examples with each feature taking value in the range $[-2.5, 2.5]$. We generate its adversarial versions by flipping the labels of farthest points from the decision boundary of a linear classifier trained on the clean dataset, using label poisoning levels (%) of $\{5, 10, 15, 20, 25\}$. We provide the visualizations of the Moon training dataset with clean and adversarial labels in Figure 5.
- IMDB review sentiment analysis benchmark dataset (Maas et al., 2011) contains train and test datasets, each containing $25,000$ examples. We used randomly selected $20,000$ points from the training set as training examples, and the rest as validation examples. We fine-tuned the RoBERTa-base model [1] (Liu et al., 2019) on this dataset and extracted features (768-dimensional embeddings) to train SVM-related models on this dataset. We generated adversarially labelled datasets using the fine-tuned RoBERTa-base model on the clean dataset. Specifically, we identified the most influential training points based on the gradient of loss with respect to the inputs and flipped their labels under various poisoning levels (%) of $\{10, 25, 30, 35, 40\}$.

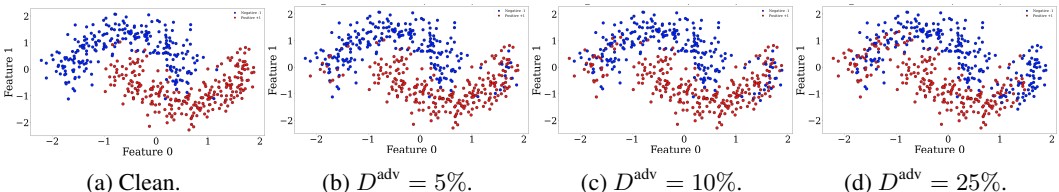

(a) Clean.      (b) $D^{\text{adv}} = 5\%$.      (c) $D^{\text{adv}} = 10\%$.      (d) $D^{\text{adv}} = 25\%$.

Figure 5: Illustrations of the Moon training sets from an example replication, using clean and adversarial labels with poisoning levels: 5%, 10%, 25%.

---

[1]The pre-trained RoBERTa-base model can be found in `https://huggingface.co/FacebookAI/roberta-base`.

## C.2 BASELINES

In our main experiments, we compared FLORAL against the baseline methods by carefully selecting their hyperparameters using the domain knowledge, which we detail below.

- LN-SVM (Biggio et al., 2011) applies a heuristic-based kernel matrix correction by assuming that every label in the training set is independently flipped with the same probability. It requires a predefined noise parameter $\mu$, which we set to $\mu \in \{0.05, 0.1, 0.15, 0.2, 0.25\}$ by leveraging the domain label poisoning knowledge, i.e. using the poisoning levels of the adversarial datasets.

- For Curie (Laishram & Phoha, 2016), we set the confidence parameter to $\{0.95, 0.90, 0.85, 0.8, 0.75\}$. To compute the average distance, we considered $k = 20$ neighbors in the same cluster for the Moon dataset and $k = 1000$ neighbors for the IMDB dataset experiments.

- For LS-SVM (Paudice et al., 2018), we use the relabeling confidence threshold from $\{0.95, 0.90, 0.85, 0.8, 0.75\}$, again aligning with the poisoning level of the adversarial datasets. For its $k$-NN step, we considered $k = 20$ and $k = 1000$ neighbors for the Moon and IMDB datasets, respectively.

- NN baseline is a fully connected multi-layer perceptron with two hidden layers with 32 units each, trained using the SGD optimizer with 0.9 momentum and binary cross-entropy loss. For additional experiments on the MNIST dataset, a similar architecture with two hidden layers having $\{32, 10\}$ units is employed.

- NN-PGD is based on the same NN architecture as above, trained with PGD-AT (Madry et al., 2017) using a standard perturbation budget of $8/255$ and a step size of $2/255$.

## C.3 ROBERTA EXPERIMENT DETAILS

We fine-tune the RoBERTa-base model[1] on the IMDB review sentiment analysis dataset[2]. We fine-tune the model for three epochs with no warm-up steps, using the AdamW optimizer, weight decay 0.01, batch size 16, and learning rate $2e-05$ with a linear scheduler, using a single NVIDIA A100 40GB GPU. We extract the last layer embeddings of the trained model for experiments with FLORAL integration.

## D EFFECTIVENESS ANALYSIS OF FLORAL DEFENSE

As explained in Section 3, FLORAL takes a *proactive* defense when the initial training data is clean, iteratively adjusting the model to reduce sensitivity to potential label poisoning attacks by exposing it to adversarial decision boundary configurations through adversarial training. Conversely, when the training data is already contaminated with adversarial labels, FLORAL mitigates their effect by *implicitly sanitizing* the corrupted labels.

To demonstrate how FLORAL defenses under already poisoned training data, we further analyze the efficacy of FLORAL by measuring its "recovery" rate of poisoned labels. That is, we quantify FLORAL's rate of disrupting the initial attack (%) on the adversarially labelled training sets, averaged over replications.

As reported in Figure 6a on the adversarial Moon datasets, FLORAL is able to disrupt the initial label attack (already inherited in the training set), at a $25 - 35\%$ rate. This contributes to the success of the FLORAL in achieving higher robust accuracy in training with adversarial datasets. Moreover, we provide example illustrations (Figures 6b-6d) that show which poisoned data points are recovered by FLORAL under the randomized top-$k$ attack.

---

[2]The IMDB review sentiment dataset can be found in `https://huggingface.co/datasets/stanfordnlp/imdb`.

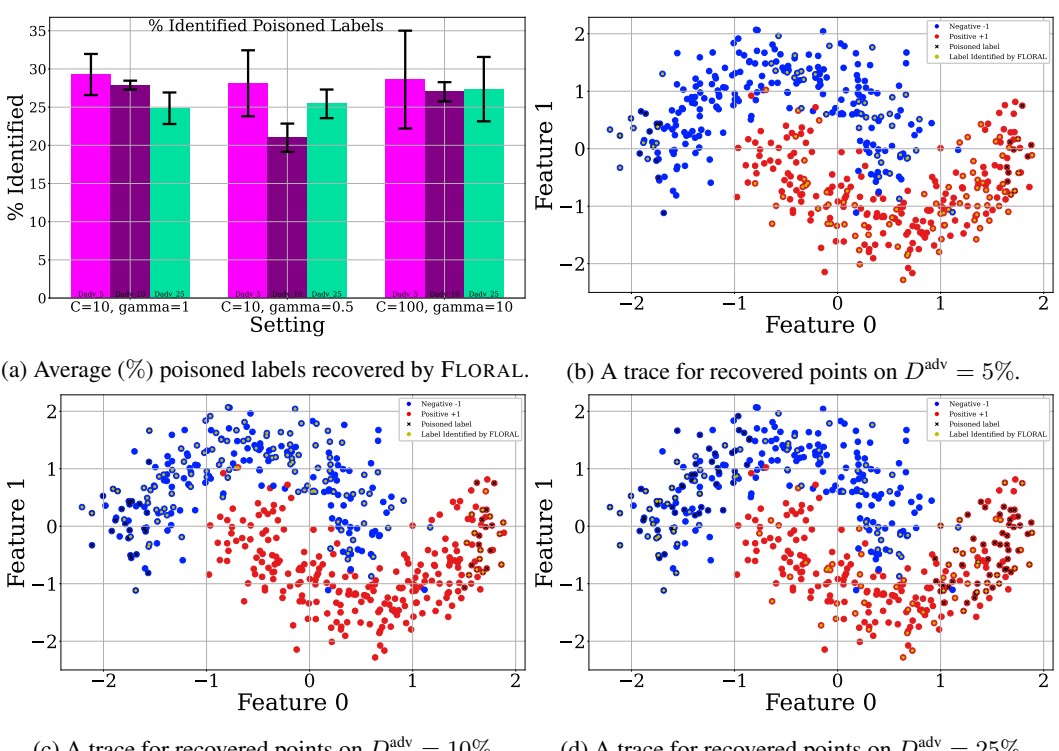

(a) Average (%) poisoned labels recovered by FLORAL.

(b) A trace for recovered points on $D^{\text{adv}} = 5\%$.

(c) A trace for recovered points on $D^{\text{adv}} = 10\%$.

(d) A trace for recovered points on $D^{\text{adv}} = 25\%$.

Figure 6: The average percentage of "**recovered**" poisoned labels by FLORAL over the adversarial Moon datasets containing $\{5, 10, 25\}$ (%) poisoned labels. As shown in **(a)**, FLORAL is able to recover, on average, $25 - 35\%$ of the poisoned labels. The plots **(b)-(d)** illustrate example traces, showing which poisoned data points are recovered by FLORAL.

# E   ADDITIONAL EXPERIMENTAL RESULTS

We provide additional experimental results under various hyperparameter settings for the Moon dataset in Appendix E.1. In Appendix E.2, we first report a comprehensive comparison of FLORAL against other baselines on the IMDB dataset, followed by an analysis of how FLORAL shifts the most influential training points for RoBERTa's predictions on the IMDB dataset. In Appendix E.3, we provide experiments on the MNIST (Deng, 2012) dataset. Finally, we present a sensitivity analysis with respect to the attacker's budget in Appendix E.4.

## E.1   MOON

We report the clean and robust test accuracy of methods under different (non-optimal) kernel hyper-parameter choices and considering label poisoning levels $\{5, 10, 25\}(\%)$ in Figure 7 and Table 4.

When the kernel hyperparameters are not optimally chosen, NN-PGD shows superior performance in less adversarial scenarios compared to SVM-based methods. However, it also demonstrates significant sensitivity to label attacks in $25\%$ adversarial settings, against all other baselines. FLORAL particularly advances by maintaining a higher robust accuracy in more adversarial settings.

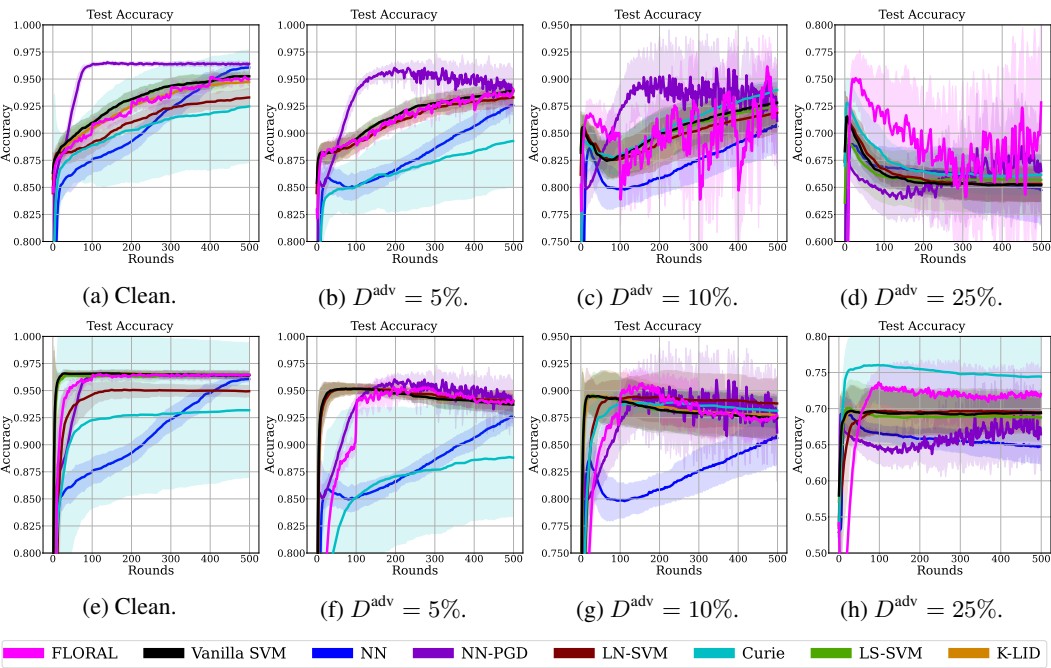

Figure 7: Comparison of clean and robust test accuracy of methods trained on the Moon dataset under different kernel hyperparameter choices. For all SVM-related models, the first row corresponds to the setting ($C = 10$, $\gamma = 0.5$), whereas the second row shows the setting ($C = 100$, $\gamma = 10$). As the level of label poisoning increases, the accuracy of models trained on adversarial datasets generally declines. When the kernel parameters are not optimally chosen, FLORAL demonstrates superior performance, particularly under the $25\%$ attack.

## E.2   IMDB

We report the test accuracy and loss performance of FLORAL against RoBERTa on the IMDB dataset in Figure 8 and Table 2. As demonstrated, FLORAL consistently exhibits superior accuracy and a smaller loss in more adversarial problem instances, without sacrificing the clean performance. This shows the effectiveness of FLORAL in achieving robust classifiers when integrated with foundation models such as RoBERTa.

Table 4: Test accuracies of methods trained on the Moon dataset. Each entry shows an average of five runs. Highlighted values indicate the best performance in the **"Best"** (peak accuracy during training) and **"Last"** (final accuracy after training) columns.

| | | Method | | | | | | | | | | | | | | |
|---|---|---|---|---|---|---|---|---|---|---|---|---|---|---|---|---|
| Setting | | FLORAL | | SVM | | NN | | NN-PGD | | LN-SVM | | Curie | | LS-SVM | | K-LID |
| | | Best | Last | Best | Last | Best | Last | Best | Last | Best | Last | Best | Last | Best | Last | Best | Last |
| Clean | $C=10, \gamma=0.5$ | 0.954 | 0.950 | 0.952 | 0.952 | 0.960 | 0.960 | **0.966** | 0.964 | 0.933 | 0.933 | 0.924 | 0.924 | 0.952 | 0.952 | 0.947 | 0.947 |
| $D^{\text{adv}}=5\%$ | $C=10, \gamma=0.5$ | 0.941 | 0.941 | 0.938 | 0.938 | 0.926 | 0.926 | **0.964** | 0.937 | 0.933 | 0.933 | 0.892 | 0.892 | 0.938 | 0.938 | 0.933 | 0.933 |
| $D^{\text{adv}}=10\%$ | $C=10, \gamma=0.5$ | 0.915 | 0.874 | 0.878 | 0.878 | 0.859 | 0.855 | **0.927** | 0.853 | 0.868 | 0.868 | 0.889 | 0.889 | 0.874 | 0.874 | 0.868 | 0.868 |
| $D^{\text{adv}}=25\%$ | $C=10, \gamma=0.5$ | **0.769** | 0.738 | 0.717 | 0.651 | 0.693 | 0.647 | 0.740 | 0.655 | 0.717 | 0.653 | 0.731 | 0.661 | 0.696 | 0.656 | 0.717 | 0.653 |
| Clean | $C=10, \gamma=1$ | **0.968** | 0.966 | 0.968 | 0.968 | 0.960 | 0.960 | 0.966 | 0.964 | 0.940 | 0.940 | 0.941 | 0.941 | 0.881 | 0.881 | 0.966 | 0.966 |
| $D^{\text{adv}}=5\%$ | $C=10, \gamma=1$ | **0.966** | 0.966 | 0.965 | 0.957 | 0.926 | 0.926 | 0.964 | 0.937 | 0.940 | 0.940 | 0.903 | 0.903 | 0.881 | 0.881 | 0.964 | 0.964 |
| $D^{\text{adv}}=10\%$ | $C=10, \gamma=1$ | 0.924 | 0.907 | 0.912 | 0.900 | 0.859 | 0.855 | **0.927** | 0.853 | 0.869 | 0.868 | 0.907 | 0.907 | 0.894 | 0.894 | 0.908 | 0.907 |
| $D^{\text{adv}}=25\%$ | $C=10, \gamma=1$ | **0.801** | 0.768 | 0.753 | 0.717 | 0.693 | 0.647 | 0.740 | 0.655 | 0.754 | 0.693 | 0.779 | 0.697 | 0.766 | 0.721 | 0.747 | 0.690 |
| Clean | $C=100, \gamma=10$ | 0.965 | 0.964 | **0.966** | 0.964 | 0.960 | 0.960 | 0.966 | 0.964 | 0.950 | 0.949 | 0.932 | 0.931 | 0.964 | 0.964 | 0.966 | 0.964 |
| $D^{\text{adv}}=5\%$ | $C=100, \gamma=10$ | 0.955 | 0.940 | 0.951 | 0.937 | 0.926 | 0.926 | **0.964** | 0.937 | 0.951 | 0.940 | 0.888 | 0.888 | 0.947 | 0.945 | 0.951 | 0.940 |
| $D^{\text{adv}}=10\%$ | $C=100, \gamma=10$ | 0.910 | 0.877 | 0.895 | 0.874 | 0.859 | 0.855 | **0.927** | 0.853 | 0.894 | 0.888 | 0.889 | 0.881 | 0.896 | 0.875 | 0.895 | 0.876 |
| $D^{\text{adv}}=25\%$ | $C=100, \gamma=10$ | 0.740 | 0.720 | 0.697 | 0.693 | 0.693 | 0.647 | 0.740 | 0.655 | 0.697 | 0.694 | **0.760** | 0.744 | 0.701 | 0.687 | 0.697 | 0.693 |

Table 5: Test accuracy and loss of methods trained on the IMDB dataset. Each entry shows an average of five replications. Highlighted values indicate the best performance in the **"Best"** (peak accuracy during training) and **"Last"** (final accuracy after training) columns. FLORAL demonstrates superior robust accuracy and lower test loss compared to RoBERTa, particularly in more adversarial scenarios.

| | Accuracy | | | | Loss | | | |
|---|---|---|---|---|---|---|---|---|
| Setting | FLORAL | | RoBERTa | | FLORAL | | RoBERTa | |
| | Best | Last | Best | Last | Best | Last | Best | Last |
| Clean | 0.911 | 0.911 | **0.911** | 0.911 | **0.196** | 0.216 | 0.229 | 0.282 |
| $D^{\text{adv}}=10\%$ | 0.903 | 0.903 | **0.904** | 0.903 | 0.234 | 0.259 | **0.227** | 0.231 |
| $D^{\text{adv}}=25\%$ | **0.889** | 0.889 | 0.882 | 0.861 | **0.310** | 0.333 | 0.337 | 0.365 |
| $D^{\text{adv}}=30\%$ | **0.880** | 0.880 | 0.867 | 0.835 | **0.353** | 0.366 | 0.428 | 0.428 |
| $D^{\text{adv}}=35\%$ | **0.871** | 0.871 | 0.827 | 0.805 | **0.381** | 0.395 | 0.496 | 0.496 |
| $D^{\text{adv}}=40\%$ | **0.863** | 0.863 | 0.779 | 0.771 | **0.428** | 0.439 | 0.551 | 0.551 |

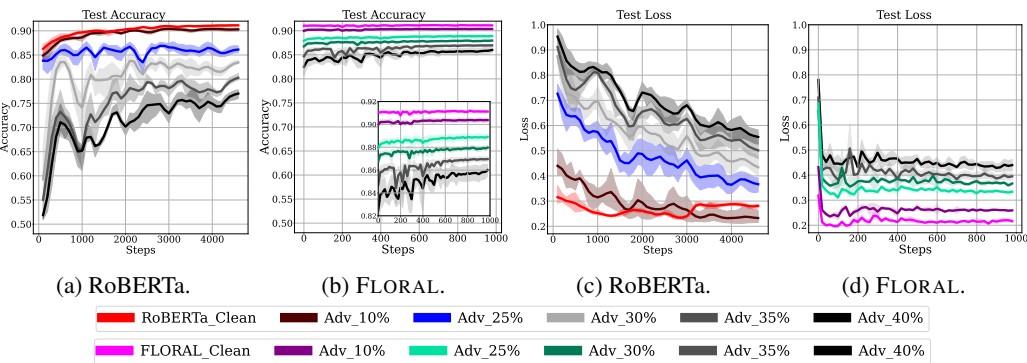

(a) RoBERTa.  (b) FLORAL.  (c) RoBERTa.  (d) FLORAL.

Figure 8: Test accuracy (**(a)-(b)**) and test loss (**(c)-(d)**) of methods on the IMDB dataset. FLORAL integration outperforms fine-tuned RoBERTa in maintaining better test accuracy and converging faster to lower loss, even when trained on extracted embeddings with heavily adversarial labels.

**Analysis on influential training points.** We further analyze how the influential training points (affecting the model's predictions) identified by FLORAL and RoBERTa change.

To illustrate, Figure 10 shows an example from a replication where both models are trained on a dataset with 40% adversarially labelled examples. We also provide the result for RoBERTa fine-tuned on the clean dataset. For FLORAL, the most influential points are selected from the most important support vectors, while for RoBERTa, these are the points yielding the largest loss gradient with respect to the input. The example clearly demonstrates that FLORAL, implemented on RoBERTa-extracted embeddings, shifts the most important training point for the model's decision boundary. FLORAL identified a more descriptive point compared to others as given in Figure 10, however, further analysis is required to determine whether FLORAL consistently identifies such training points across all cases.

Additionally, we investigate the overlap in influential training points between the two methods. To this end, for each method, we extract the $25\%$ most influential training points (for the model predictions) among the training dataset, and measure how much overlap between these two sets. In Figure 9, we report the percentage of "common" influential points identified from the IMDB dataset, averaged over replications, with error bars denoting the standard deviation. The left figure shows the percentage overlap between FLORAL trained on the IMDB dataset with different poison levels and RoBERTa fine-tuned on clean labels. Whereas, the right plot shows the overlap between both models trained on the dataset with different poison levels. On the clean dataset, although there are some differences, both methods almost identify the same set of influential points. However, as adversarial labels increase, the overlap decreases. This shows that FLORAL extracts more critical points that enhance the model's robustness in adversarial settings, as supported by its superior robust accuracy, already shown in Figure 8 in Section 4.1.

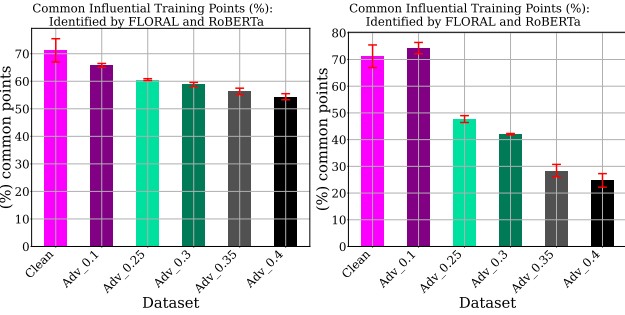

(a) w.r.t. RoBERTa (clean).    (b) w.r.t. RoBERTa (adversarial).

Figure 9: The percentage of "common" influential points identified by FLORAL and RoBERTa from the IMDB dataset, averaged over replications, with error bars denoting the standard deviation. "Clean" shows the dataset with clean labels, whereas adversarial datasets contain $\{10, 25, 30, 35, 40\}(\%)$ poisoned labels. Even when both methods are fine-tuned on the clean dataset, slight differences emerge in the identified influential training points. The discrepancy increases as the dataset becomes more adversarial, highlighting that FLORAL adjusts the influential training points affecting the model's predictions.

**FLORAL(Dadv=40%):**

Sitting down to watch the 14th season of the Bachelor ("On the Wings of Love"), I knew I would be in for an "interesting" time. I had watched some of the previous seasons of the Bachelor in passing; watching an episode or two and missing the next three or so. I find that the Bachelor is often appealing and intriguing, though its quality and morality are often lacking.

"On the Wings of Love" details the journey taken by Jake, a 31 year old commercial pilot from Dallas, Texas, to find true love, as true a love as one can find in a season-long reality-drama dating show. Jake meets 25 beautiful girls from all over the country. He begins to get to know them a bit, but it is mostly superficial; how well can you get to know someone in a few 5 minute conversations? Jake tries to make his true intentions known from the very beginning, at least to the audience. He noted that he doesn't just want love or a good time, but he wants a fiancé or wife. We can only assume that he has made this clear to the women in the competition as well. If that is the case, it might explain, to a degree, some of the women's actions. The women are super competitive. While they don't even know Jake at all yet, they are still in it to win it no matter what the cost.

Not only were the women competitive, but they were also confident and catty. Threats, backstabbing, and warnings of "Watch out!" all show that these women weren't there for a good time either. Jake noted that he was not just looking for sex appeal, but looking for "a connection." However, the girls pulled out all the stops to try to impress Jake with said sex appeal. They arrived at the mansion in skimpy dresses —either low-cut or short.

While some girls seemed to maintain their sense of decorum, others missed that memo altogether. One girl, Channy, noted that Jake was a "good guy" to whom she could be a "naughty girl." She went on to say that Jake could land on her "runway anytime." She got flack from the other girls for her provocative statement which showed their take on these situations.

So, a reality dating show couldn't be that bad, could it? Besides the obvious issue of sex-driven attraction, there are other issues that mar this seemingly harmless show. Is this the right way to find a future mate; vying for someone's attention by flaunting oneself to extreme proportions? Unfortunately, however, that is what America has reduced dating to these days: pleasure and sex without commitment and a little happiness on the side.

Another problem is the premature emotional attachment by which many of the girls bound themselves to Jake. A few girls in particular seemed to be overly attached. One girl said "If I don't get that first impression rose it will kill me!" As mentioned before, they don't even know him yet and she was talking about a specific rose, not just one of the 15 roses to keep from being eliminated.

Michelle, in particular, seemed to have some issues with attachment to Jake. The other girls noticed it too. After one particular Michelle outburst, Vienna asserted that Michelle had a "mental breakdown and we've only been here an hour." Michelle got the last rose of the evening on the first show —narrowly missing elimination —and was extremely emotional about it. The other girls thought it was simply ridiculous. Another girl also cried, but because she was eliminated.

It began with Survivor, and from there it just took off —reality TV. It shows our entertainment interests as a country; if we weren't watching the shows and giving them good ratings, the networks would not continue to run them. The only logical conclusion that can be drawn is that enough of America is hooked. One thing is clear: America (in general) loves reality TV and its ensuing trappings.

This begs me to question: why is it that we even like reality TV? What is it about it that draws us to it? Is it because we see the similarities to our own lives, or is it because we want to be sure that we are more stable and less pathetic than others? Whatever it is that draws us to it, we should be careful of the media and entertainment that we allow to fill our minds. I'm not saying that all reality TV shows are bad; however, I am saying that we need to evaluate each one.

Episodes used for critique: Season Premier and Episode 2.

**(negative)**

**RoBERTa(fine-tuned on clean):**

This documentary on schlockmeister William Castle takes a few cheap shots at the naive '50s-'60s environment in which he did his most characteristic work--look at the funny, silly people with the ghost-glasses--but it's also affectionate and lively, with particularly bright commentary from John Waters, who was absolutely the target audience for such things at the time, and from Castle's daughter, who adored her dad and also is pretty perceptive about how he plied his craft. (We never find out what became of the other Castle offspring.) The movies were not very good, it makes clear, but his marketing of them was brilliant, and he appears to have been a sweet, hardworking family man. Fun people keep popping up, like "Straight Jacket"'s Diane Baker, who looks great, and Anne Helm, whom she replaced at the instigation of star Joan Crawford. Darryl Hickman all but explodes into giggles at the happy memory of working with Castle on "The Tingler," and there's enough footage to give us an idea of the level of Castle's talent--not very high, but very energetic. A pleasant look at a time when audiences were more easily pleased, and it does make you nostalgic for simpler movie-going days.

**(positive)**

**RoBERTa(fine-tuned on Dadv=40%):**

Someone actually gave this movie 2 stars. There's a very high chance they need immediate professional help as anyone who doesn't spend 30 seconds to see if you can award no stars is quite literally scary.

This film is ... well ... I guess it's pretty much some kind of attempt at a horrible porn / snuff movie with no porn or no real horrible bits (apart from the acting, plot, story, sets, dialogue and sound). I wrongly assumed it was about zombies. 

Watching it is actually quite scary in fairness; you're terrified someone will come over and you'll never be able to describe what it is and they'll go away thinking you're a freak that watches home-made amateur torture videos or something along those lines. 

I'm so taken aback I'm writing this review on my mobile so I don't forget to attempt to bring the rating down further than the current 1.6 to save others from the same horrible fate that I just suffered. 

I worst film I've ever seen and I can say (with hand on heart) it will never, never be topped.

**(negative)**

Figure 10: *Note: Figure might contain offensive content.* The most influential training point for the model's predictions, identified by FLORAL and RoBERTa from the IMDB dataset. FLORAL implemented on RoBERTa extracted-embeddings changes the most important training point for the model's decision boundary.

### E.3 MNIST

To demonstrate the generalizability of FLORAL across diverse datasets, we provide additional experiments on the MNIST dataset (Deng, 2012). Similar to (Rosenfeld et al., 2020), we consider classes 1 and 7 which leads to a dataset of $\mathcal{D} = \{(x_i, y_i)\}_{i=1}^{13007}$ where $x_i \in \mathbb{R}^{784}$ and $y_i \in \{\pm 1\}$, with 784 pixel values for each image.

We perform the randomized top-$k$ label poisoning attack described in Section 3 and report the clean and robust test accuracy performance of methods in Figure 11 and Table 6. The results show that FLORAL maintains a higher robust accuracy compared to most of the baselines. While Curie behaves almost on par, FLORAL achieves higher robust accuracy. Although NN baselines perform better on clean and 5% adversarially labelled datasets, they show a significant accuracy decrease when the training dataset gets more adversarial.

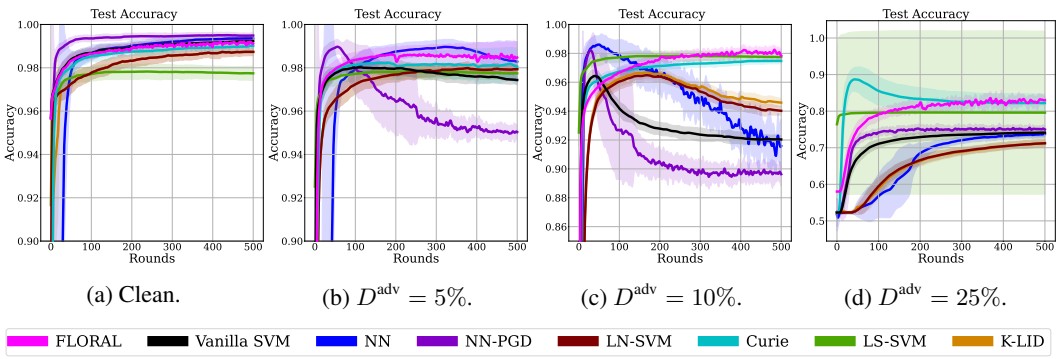

|     (a) Clean.     |     (b) $D^{\mathrm{adv}} = 5\%$.     |     (c) $D^{\mathrm{adv}} = 10\%$.     |     (d) $D^{\mathrm{adv}} = 25\%$.     |

FLORAL  Vanilla SVM  NN  NN-PGD  LN-SVM  Curie  LS-SVM  K-LID

Figure 11: Clean and robust test accuracy of methods trained on the MNIST-1vs7 dataset. "Clean" refers to the dataset with clean labels, while the adversarial datasets contain $\{5, 10, 25\}$ (%) poisoned labels. For all SVM-related models, the setting $C = 5$, $\gamma = 0.005$ is used. As the level of label poisoning increases, models trained on adversarial datasets generally demonstrate a decline in accuracy. However, FLORAL maintains a higher robust accuracy level compared to most of the baselines and behaving on par with the Curie method.

Table 6: Test accuracies of methods trained over the MNIST-1vs7 dataset. Each entry shows the average of five replications with different train/test splits. Highlighted values indicate the best performance in the **"Best"** (peak accuracy during training) and **"Last"** (final accuracy after training) columns.

| | | \multicolumn{16}{c}{Method} | | | | | | | | | | | | | | |
| | | \multicolumn{2}{c}{FLORAL} | \multicolumn{2}{c}{SVM} | \multicolumn{2}{c}{NN} | \multicolumn{2}{c}{NN-PGD} | \multicolumn{2}{c}{LN-SVM} | \multicolumn{2}{c}{Curie} | \multicolumn{2}{c}{LS-SVM} | \multicolumn{2}{c}{K-LID} |
| \multicolumn{2}{c}{Setting} | Best | Last | Best | Last | Best | Last | Best | Last | Best | Last | Best | Last | Best | Last | Best | Last |
|---|---|---|---|---|---|---|---|---|---|---|---|---|---|---|---|---|---|---|
| Clean | $C=5, \gamma=0.005$ | 0.992 | 0.991 | 0.992 | 0.992 | 0.993 | 0.993 | **0.995** | 0.994 | 0.987 | 0.987 | 0.990 | 0.990 | 0.978 | 0.977 | 0.987 | 0.987 |
| $D^{\mathrm{adv}}=5\%$ | $C=5, \gamma=0.005$ | 0.988 | 0.984 | 0.980 | 0.974 | 0.989 | 0.982 | 0.989 | 0.949 | 0.979 | 0.979 | 0.984 | 0.979 | 0.978 | 0.977 | 0.979 | 0.979 |
| $D^{\mathrm{adv}}=10\%$ | $C=5, \gamma=0.005$ | **0.984** | 0.978 | 0.964 | 0.920 | 0.982 | 0.930 | 0.982 | 0.894 | 0.965 | 0.940 | 0.974 | 0.974 | 0.978 | 0.977 | 0.966 | 0.945 |
| $D^{\mathrm{adv}}=25\%$ | $C=5, \gamma=0.005$ | 0.853 | 0.830 | 0.741 | 0.741 | 0.738 | 0.738 | 0.763 | 0.750 | 0.712 | 0.712 | **0.887** | 0.822 | 0.796 | 0.795 | 0.712 | 0.712 |

### E.4 SENSITIVITY ANALYSIS

In our experiments with the Moon dataset under varying label poisoning levels, we consider attacker budgets $B = 2k$ under varying $k$ values, and report the best performing setting in Figure 3 in Section 4.1.

However, we further investigate the sensitivity of FLORAL to the attacker's budget $B$, by considering levels $B \in \{5, 10, 25, 50, 125\}$, with results presented in Figure 12. As demonstrated, FLORAL shows superior performance under a constrained attacker budget in the clean label scenario, as expected, since an increasing number of adversarially labelled examples during training degrades clean test accuracy. In contrast, baseline methods operate on a fixed dataset. However, as the dataset gets more adversarial, FLORAL outperforms under higher attacker budgets.

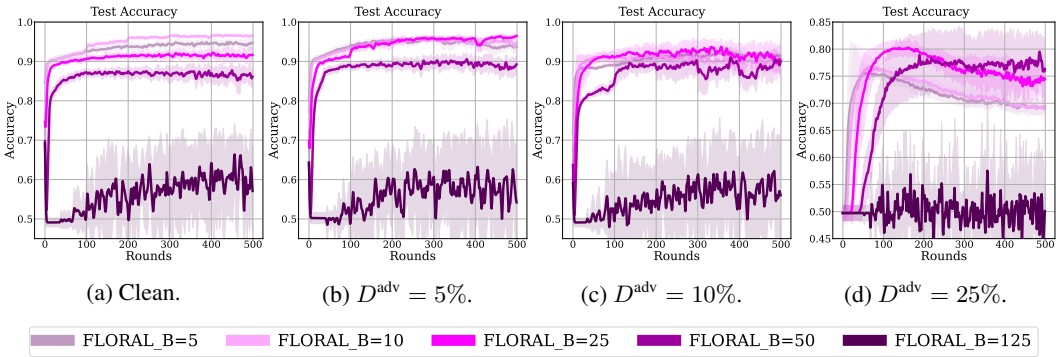

Figure 12: The sensitivity of FLORAL to the attacker's budget $B$. "Clean" refers to the dataset with clean labels, while the adversarial datasets contain $\{5, 10, 25\}(\%)$ poisoned labels. The performance under setting ($C = 10$, $\gamma = 1$) is presented. As the level of label poisoning increases, FLORAL performs better under higher attacker budget settings.

## F EXTENSION TO MULTI-CLASS CLASSIFICATION

We extend our algorithm to multi-class classification tasks, as detailed in Algorithm 3. The primary modification involves adopting a one-vs-all approach (Hsu & Lin, 2002) by employing kernel SVM model $f_{\lambda_0}^m$ for each class $m \in \mathcal{M}$ and associating multiple attackers $a_m, m \in \mathcal{M}$ for the corresponding classifiers. In each round $t$, the attackers identify the $B_m$ most influential data points with respect to $\lambda_t^m$ values of the corresponding models under their constrained budgets $B_m$, and gather them into a set $\mathcal{B}_t$. Among the points in $\mathcal{B}_t$, the labels of top-$k$ influential data points are poisoned according to a predefined label poisoning distribution $q$. The dataset with adversarial labels is then shared with each kernel SVM model and local training is applied via PGD training step.

---

**Algorithm 3** FLORAL-MultiClass

1: **Input:** Initial kernel SVM models $f_{\lambda_0}^m$ for each class $m \in \mathcal{M}$, training dataset $\mathcal{D}_0 = \{(x_i, y_i)\}_{i=1}^n, x_i \in \mathbb{R}^d, y_i \in \{\pm 1\}$, attackers' budgets $B_m$, parameter $k$, where $k \ll \min\{B_m\}_{m \in \mathcal{M}}$, learning rate $\eta$, a pre-defined label flip distribution $q$.
2: **for** round $t = 1, \ldots, T$ **do**
3:     Initialize $\mathcal{B}_t \leftarrow \emptyset$.
4:     **for** $m \in \mathcal{M}$ **do**
5:         $\mathcal{B}_t^m \leftarrow$ Identify top-$B_m$ support vectors w.r.t. $\lambda_{t-1}^m$ values by solving (7-9).
6:     **end for**
7:     $\mathcal{B}_t \leftarrow \bigcup_{m \in \mathcal{M}} \mathcal{B}_t^m$.
8:     $\tilde{y}^t \leftarrow$ randomized top-$k$ : Randomly select $k$ points among $\mathcal{B}_t$ and poison their labels w.r.t. $q$.
9:     $\mathcal{D}_t \leftarrow \{(x_i, \tilde{y}_i^t)\}_{i=1}^n$         */ Adversarial dataset with selected $k$ poisoned labels
10:     **for** $m \in \mathcal{M}$ **do**
11:         Compute gradient of the objective (4), $\nabla_\lambda D(f_{\lambda_{t-1}}^m; \mathcal{D}_t)$, based on $\lambda_{t-1}^m, \mathcal{D}_t$ as given in (10).
12:         Take a PGD step $\lambda_t^m \leftarrow$ PROX $_{\mathcal{S}(\tilde{y}^t)}(\lambda_{t-1}^m - \eta \nabla_\lambda D(f_{\lambda_{t-1}}^m; \mathcal{D}_t))$.   */ Adversarial training
13:     **end for**
14: **end for**
15: **return** $\{f_{\lambda_T}^m\}_{m \in \mathcal{M}}$

---

## G COMPARISON AGAINST ADDITIONAL METHODS

We additionally compare FLORAL against least squares classifier using randomized smoothing (RS) (Rosenfeld et al., 2020), and regularized synthetic reduced nearest neighbor (RSRNN) (Tavallali et al., 2022) methods on the Moon and MNIST-1vs7 datasets. RS provides a robustness certification for a linear classifier under label-flipping attacks. Whereas, RSRNN, conceptually similar to Curie (Laishram & Phoha, 2016), provides a filtering-out defense based on clustering.

We evaluated the performance under different noise ($q$) and $l_2$ regularization ($\lambda$) hyperparameter values for the RS method suggested in (Rosenfeld et al., 2020), whereas we considered varying number of centroids $K$, penalty coefficient $\lambda$, cost complexity coefficient $\alpha$, for the RSRNN method, using referenced values in the study (Tavallali et al., 2022) for the MNIST dataset.

The results presented in Tables 7-10 demonstrate that FLORAL consistently outperforms both RS and RSRNN across all datasets and experimental settings. While RSRNN achieves comparable performance on the MNIST dataset, it still falls short of FLORAL. The performance of the RS method, which employs a linear classifier with a pointwise robustness certificate, aligns with expectations, as its simpler classifier limits its ability to capture complex patterns. In contrast, FLORAL utilizes kernel SVMs, enabling it to effectively model intricate patterns within the data and achieve superior results.

Table 7: Test accuracies of FLORAL against randomized smoothing (RS) method (Rosenfeld et al., 2020) on the Moon dataset. Each entry shows an average of five runs. We evaluated the performance under different noise ($q$) values for RS.

| Setting | | Method | | | |
|---|---|---|---|---|---|
| | | FLORAL | RS ($q = 0.1, \lambda = 0.01$) | RS ($q = 0.3, \lambda = 0.01$) | RS ($q = 0.4, \lambda = 0.01$) |
| Clean | $C = 10, \gamma = 1$ | **0.968** | 0.557 | 0.509 | 0.509 |
| $D^{adv} = 5\%$ | $C = 10, \gamma = 1$ | **0.963** | 0.552 | 0.509 | 0.509 |
| $D^{adv} = 10\%$ | $C = 10, \gamma = 1$ | **0.954** | 0.540 | 0.509 | 0.509 |
| $D^{adv} = 25\%$ | $C = 10, \gamma = 1$ | **0.776** | 0.520 | 0.505 | 0.505 |

Table 8: Test accuracies of FLORAL against randomized smoothing (RS) method (Rosenfeld et al., 2020) on the MNIST-1vs7 dataset. Each entry shows an average of five runs. We evaluated the performance under different noise ($q$) and $l_2$ regularization ($\lambda$) hyperparameter values for RS, as suggested in (Rosenfeld et al., 2020).

| Setting | | Method | | | | | | |
|---|---|---|---|---|---|---|---|---|
| | | FLORAL | RS ($q = 0.1, \lambda = 0.01$) | RS ($q = 0.3, \lambda = 0.01$) | RS ($q = 0.4, \lambda = 0.01$) | RS ($q = 0.1, \lambda = 12291$) | RS ($q = 0.3, \lambda = 12291$) | RS ($q = 0.4, \lambda = 13237$) |
| Clean | $C = 5, \gamma = 0.005$ | **0.991** | 0.973 | 0.921 | 0.836 | 0.940 | 0.846 | 0.732 |
| $D^{adv} = 5\%$ | $C = 5, \gamma = 0.005$ | **0.984** | 0.921 | 0.876 | 0.800 | 0.895 | 0.802 | 0.701 |
| $D^{adv} = 10\%$ | $C = 5, \gamma = 0.005$ | **0.978** | 0.868 | 0.831 | 0.768 | 0.830 | 0.745 | 0.673 |
| $D^{adv} = 25\%$ | $C = 5, \gamma = 0.005$ | **0.830** | 0.706 | 0.693 | 0.669 | 0.548 | 0.594 | 0.595 |

Table 9: Test accuracies of FLORAL against regularized synthetic reduced nearest neighbor (RSRNN) (Tavallali et al., 2022) trained on the Moon dataset. Each entry shows an average of five runs. We evaluated the performance under different hyperparameter values (number of centroids $K$, penalty coefficient $\lambda$, cost complexity coefficient $\alpha$) for the RSRNN method.

| Setting | | Method | | | | |
|---|---|---|---|---|---|---|
| | | FLORAL | RSRNN ($K = 2, \alpha = 0.01, \lambda = 0.1$) | RSRNN ($K = 10, \alpha = 0.01, \lambda = 0.1$) | RSRNN ($K = 10, \alpha = 0.1, \lambda = 1$) | RSRNN ($K = 20, \alpha = 0.01, \lambda = 0.1$) |
| Clean | $C = 10, \gamma = 1$ | **0.968** | 0.505 | 0.629 | 0.688 | 0.617 |
| $D^{adv} = 5\%$ | $C = 10, \gamma = 1$ | **0.963** | 0.502 | 0.547 | 0.603 | 0.512 |
| $D^{adv} = 10\%$ | $C = 10, \gamma = 1$ | **0.954** | 0.502 | 0.532 | 0.566 | 0.482 |
| $D^{adv} = 25\%$ | $C = 10, \gamma = 1$ | **0.776** | 0.494 | 0.434 | 0.476 | 0.439 |

Table 10: Test accuracies of FLORAL against regularized synthetic reduced nearest neighbor (RSRNN) (Tavallali et al., 2022) trained on the MNIST-1vs7 dataset. Each entry shows an average of five runs. We evaluated the performance under different cost complexity coefficient ($\alpha$) values for the RSRNN method.

| Setting | | Method | | |
|---|---|---|---|---|
| | | FLORAL | RSRNN ($K = 10, \alpha = 0.1, \lambda = 1.0$) | RSRNN ($K = 10, \alpha = 1.0, \lambda = 1.0$) |
| Clean | $C = 5, \gamma = 0.005$ | **0.991** | 0.619 | 0.692 |
| $D^{adv} = 5\%$ | $C = 5, \gamma = 0.005$ | **0.984** | 0.599 | 0.441 |
| $D^{adv} = 10\%$ | $C = 5, \gamma = 0.005$ | **0.978** | 0.432 | 0.408 |
| $D^{adv} = 25\%$ | $C = 5, \gamma = 0.005$ | **0.830** | 0.403 | 0.408 |

# H  EXPERIMENTS UNDER DIFFERENT LABEL ATTACKS

To show the generalizability of our approach in the presence of otherlabel poisoning attack types, we further compare FLORAL against baselines on adversarial datasets generated using the `alfa`, `alfa-tilt` (Xiao et al., 2015) and LFA attacks (Paudice et al., 2018).

## H.1 EXPERIMENTS WITH THE `ALFA-TILT` ATTACK

We further evaluate FLORAL's performance in the presence of `alfa-tilt` attack (Xiao et al., 2015) on the Moon and MNIST-1vs7 datasets. We report the results on the Moon datasets in Table 11, whereas we present the results for MNIST-1vs7 dataset in Figure 13 and Table 12.

As shown with the results on the Moon dataset, FLORAL is able to achieve a higher "Best" robust accuracy level throughout the training process. Furthermore, FLORAL's effectiveness under `alfa-tilt` attack is best shown on the MNIST dataset. As reported in Figure 13 and Table 12, FLORAL achieves an outperforming robust accuracy level compared to baseline methods on all adversarial settings. This demonstrates the potential of FLORAL defense against other label poisoning attacks.

Table 11: Test accuracies of methods trained over the Moon dataset with adversarial labels generated by the `alfa-tilt` (Xiao et al., 2015) attack. Each entry shows the average of five replications. Highlighted values indicate the best performance in the "**Best**" (peak accuracy during training) and "Last" (final accuracy after training) columns.

| Setting | | Method | | | | | | | | | | | | | | | |
|---|---|---|---|---|---|---|---|---|---|---|---|---|---|---|---|---|---|
| | | FLORAL | | SVM | | NN | | NN-PGD | | LN-SVM | | Curie | | LS-SVM | | K-LID | |
| | | Best | Last | Best | Last | Best | Last | Best | Last | Best | Last | Best | Last | Best | Last | Best | Last |
| Clean | $C=10, \gamma=1$ | **0.968** | 0.966 | 0.968 | 0.968 | 0.960 | 0.960 | 0.966 | 0.964 | 0.940 | 0.940 | 0.941 | 0.941 | 0.881 | 0.881 | 0.966 | 0.966 |
| $D^{adv}=5\%$ | $C=10, \gamma=1$ | **0.972** | 0.957 | 0.944 | 0.939 | 0.948 | 0.948 | 0.962 | 0.943 | 0.956 | 0.956 | 0.940 | 0.939 | 0.898 | 0.896 | 0.937 | 0.936 |
| $D^{adv}=10\%$ | $C=10, \gamma=1$ | **0.971** | 0.928 | 0.910 | 0.886 | 0.915 | 0.914 | 0.940 | 0.906 | 0.930 | 0.930 | 0.920 | 0.902 | 0.898 | 0.896 | 0.926 | 0.926 |
| $D^{adv}=25\%$ | $C=10, \gamma=1$ | **0.893** | 0.824 | 0.787 | 0.722 | 0.837 | 0.750 | 0.837 | 0.720 | 0.786 | 0.723 | 0.792 | 0.759 | 0.792 | 0.791 | 0.770 | 0.708 |

Table 12: Test accuracies of methods trained on the MNIST-1vs7 dataset under `alfa-tilt` poisoning attack (Xiao et al., 2015). Each entry shows the average of five replications with different train/test splits. Highlighted values indicate the best performance in the "**Best**" (peak accuracy during training) and "Last" (final accuracy after training) columns.

| Setting | | Method | | | | | | | | | | | | | | | |
|---|---|---|---|---|---|---|---|---|---|---|---|---|---|---|---|---|---|
| | | FLORAL | | SVM | | NN | | NN-PGD | | LN-SVM | | Curie | | LS-SVM | | K-LID | |
| | | Best | Last | Best | Last | Best | Last | Best | Last | Best | Last | Best | Last | Best | Last | Best | Last |
| Clean | $C=5, \gamma=0.005$ | 0.992 | 0.991 | 0.992 | 0.992 | 0.993 | 0.993 | **0.995** | 0.994 | 0.987 | 0.987 | 0.990 | 0.990 | 0.978 | 0.977 | 0.987 | 0.987 |
| $D^{adv}=5\%$ | $C=5, \gamma=0.005$ | **0.991** | 0.990 | 0.980 | 0.980 | 0.991 | 0.958 | 0.988 | 0.955 | 0.979 | 0.979 | 0.987 | 0.987 | 0.980 | 0.979 | 0.978 | 0.978 |
| $D^{adv}=10\%$ | $C=5, \gamma=0.005$ | 0.984 | 0.982 | 0.970 | 0.970 | 0.986 | 0.917 | **0.988** | 0.909 | 0.966 | 0.966 | 0.974 | 0.974 | 0.979 | 0.978 | 0.965 | 0.965 |
| $D^{adv}=25\%$ | $C=5, \gamma=0.005$ | 0.811 | 0.788 | 0.713 | 0.713 | 0.795 | 0.739 | **0.824** | 0.754 | 0.703 | 0.701 | 0.734 | 0.734 | 0.526 | 0.526 | 0.707 | 0.705 |

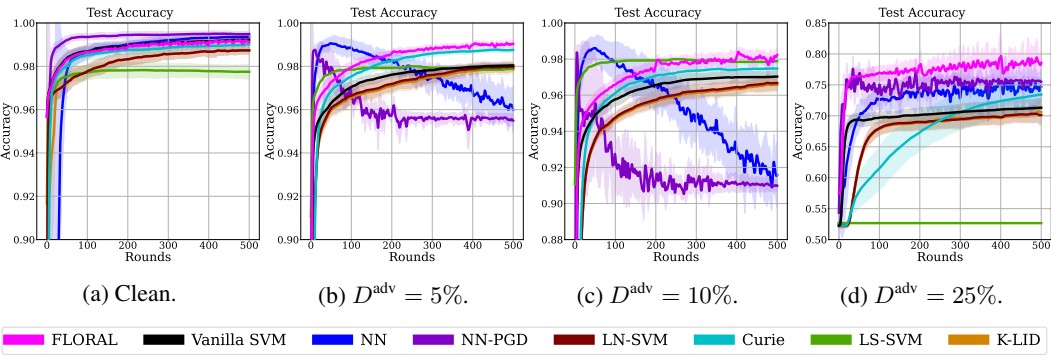

(a) Clean.     (b) $D^{adv}=5\%$.     (c) $D^{adv}=10\%$.     (d) $D^{adv}=25\%$.

FLORAL    Vanilla SVM    NN    NN-PGD    LN-SVM    Curie    LS-SVM    K-LID

Figure 13: Clean and robust test accuracy of methods trained on the MNIST-1vs7 dataset under `alfa-tilt` poisoning attack (Xiao et al., 2015). "Clean" refers to the dataset with clean labels, while the adversarial datasets contain $\{5, 10, 25\}$ (%) poisoned labels. For all SVM-related models, the setting $C=5$, $\gamma=0.005$ is used. FLORAL achieves outperforming robust accuracy level compared to baseline methods on all adversarial settings, clearly demonstrating the potential of FLORAL as a defense against other types of label poisoning attacks.

## H.2 EXPERIMENTS WITH THE `ALFA` ATTACK

The `alfa` attack is generated under the assumption that the attacker can maliciously alter the training labels to maximize the empirical loss of the original classifier on the tainted dataset. From this, the attacker's objective is formulated as maximizing the difference between the empirical risk for classifiers under tainted and untainted label sets.

We experimented on the Moon dataset and considered label poisoning levels (%) of $\{5, 10, 25\}$. The illustrations of the Moon dataset with clean and `alfa`-attacked adversarial labels are given Figure 14. We report the results under `alfa` attack in Figure 15 and Table 13. As shown, FLORAL is especially effective against vanilla SVM in maintaining higher robust accuracy in adversarial settings. NN-based methods again fail drastically as the dataset becomes more adversarial under `alfa` attack. Although LS-SVM shows premise in moderate adversarial scenarios, it fails to be effective considering clean and most adversarial performance. Furthermore, FLORAL demonstrates superior performance compared to all baselines in the most adversarial setting (with 25% poisoned labels).

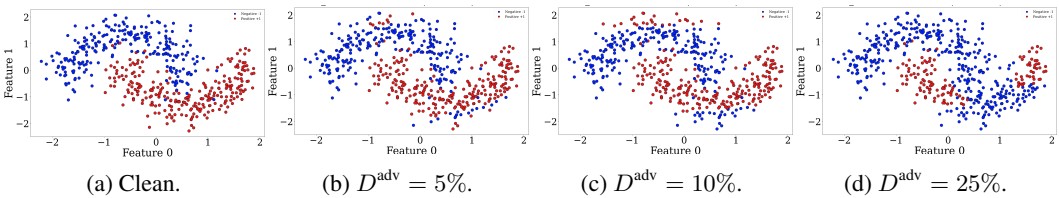

(a) Clean.    (b) $D^{\mathrm{adv}} = 5\%$.    (c) $D^{\mathrm{adv}} = 10\%$.    (d) $D^{\mathrm{adv}} = 25\%$.

Figure 14: Illustrations of the Moon training sets from an example replication, using clean and `alfa`-attacked adversarial labels with poisoning levels: 5%, 10%, 25%.

Table 13: Test accuracies of methods trained over the Moon dataset with adversarial labels generated by the `alfa` attack. Each entry shows the average of five replications with different train/test splits. Highlighted values indicate the best performance in the **"Best"** (peak accuracy during training) and "Last" (final accuracy after training) columns.

| Setting | | FLORAL | | SVM | | NN | | NN-PGD | | LN-SVM | | Curie | | LS-SVM | | K-LID | |
|---|---|---|---|---|---|---|---|---|---|---|---|---|---|---|---|---|---|
| | | Best | Last | Best | Last | Best | Last | Best | Last | Best | Last | Best | Last | Best | Last | Best | Last |
| Clean | $C=10, \gamma=1$ | **0.968** | 0.966 | 0.968 | 0.968 | 0.960 | 0.960 | 0.966 | 0.964 | 0.940 | 0.940 | 0.941 | 0.941 | 0.881 | 0.881 | 0.966 | 0.966 |
| $D^{\mathrm{adv}}=5\%$ | $C=10, \gamma=1$ | 0.963 | 0.950 | 0.954 | 0.946 | 0.875 | 0.875 | 0.963 | 0.958 | 0.942 | 0.942 | 0.934 | 0.933 | **0.964** | 0.964 | 0.942 | 0.942 |
| $D^{\mathrm{adv}}=10\%$ | $C=10, \gamma=1$ | 0.954 | 0.902 | 0.914 | 0.893 | 0.836 | 0.816 | 0.918 | 0.895 | 0.914 | 0.907 | 0.915 | 0.914 | **0.955** | 0.954 | 0.914 | 0.907 |
| $D^{\mathrm{adv}}=25\%$ | $C=10, \gamma=1$ | **0.776** | 0.763 | 0.750 | 0.750 | 0.693 | 0.658 | 0.693 | 0.645 | 0.729 | 0.729 | 0.741 | 0.741 | 0.740 | 0.740 | 0.729 | 0.729 |

## H.3 EXPERIMENTS WITH THE `LFA` ATTACK

We additionally evaluate FLORAL's effectiveness compared to baselines in the presence of `LFA` attack (Paudice et al., 2018) on the Moon dataset. As results are shown in Figure 16 and Table 14, FLORAL demonstrates significant performance when the label poisoning attack level is high, i.e., 10% or 25%. However, under those settings, LS-SVM (Paudice et al., 2018) baseline shows faster convergence, which is expected as the LS-SVM (Paudice et al., 2018) method is specifically crafted against the `LFA` attack. Considering that LS-SVM fails in clean test performance, it is clear that FLORAL provides a generalizable defense.

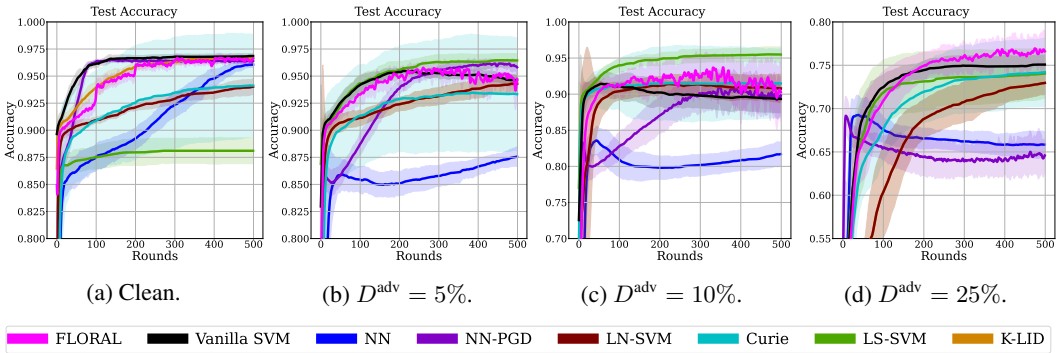

Figure 15: Clean and robust test accuracy of methods trained on the Moon dataset under `alfa` poisoning attack. "Clean" refers to the dataset with clean labels, while the adversarial datasets contain $\{5, 10, 25\}(\%)$ poisoned labels. As the level of label poisoning increases, models trained on adversarial datasets generally demonstrate a decline in accuracy. However, FLORAL demonstrates a gradually improving robust accuracy performance, particularly when the attack intensity increases to 25%.

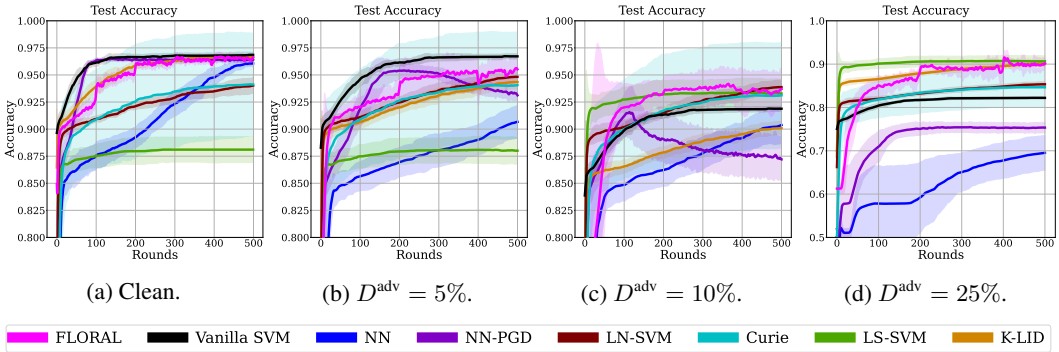

Figure 16: Clean and robust test accuracy of methods trained on the Moon dataset under `LFA` poisoning attack (Paudice et al., 2018). "Clean" refers to the dataset with clean labels, while the adversarial datasets contain $\{5, 10, 25\}$ (%) poisoned labels. For all SVM-related models, the setting $C = 1, \gamma = 1.0$ is used. As the level of label poisoning increases, models trained on adversarial datasets generally demonstrate a decline in accuracy. However, FLORAL demonstrates a gradually improving robust accuracy performance, particularly when the attack level is 10% or 25%.

Table 14: Test accuracies of methods trained over the Moon dataset with adversarial labels generated by the `LFA` (Paudice et al., 2018) attack. Each entry shows the average of five replications. Highlighted values indicate the best performance in the "**Best**" (peak accuracy during training) and "Last" (final accuracy after training) columns.

| | | | | | | | | | | | | | | | | | | |
|---|---|---|---|---|---|---|---|---|---|---|---|---|---|---|---|---|---|
| | | \multicolumn{16}{c}{**Method**} | | | | | | | | | | | | | | |
| \multirow{2}{*}{**Setting**} | | FLORAL | | SVM | | NN | | NN-PGD | | LN-SVM | | Curie | | LS-SVM | | K-LID | |
| | | Best | Last | Best | Last | Best | Last | Best | Last | Best | Last | Best | Last | Best | Last | Best | Last |
| Clean | $C=10, \gamma=1$ | **0.968** | 0.966 | 0.968 | 0.968 | 0.960 | 0.960 | 0.966 | 0.964 | 0.940 | 0.940 | 0.941 | 0.941 | 0.881 | 0.881 | 0.966 | 0.966 |
| $D^{\mathrm{adv}}=5\%$ | $C=10, \gamma=1$ | 0.957 | 0.954 | **0.967** | 0.967 | 0.906 | 0.906 | 0.955 | 0.930 | 0.948 | 0.948 | 0.940 | 0.940 | 0.880 | 0.880 | 0.943 | 0.943 |
| $D^{\mathrm{adv}}=10\%$ | $C=10, \gamma=1$ | **0.943** | 0.937 | 0.919 | 0.918 | 0.903 | 0.903 | 0.917 | 0.872 | 0.938 | 0.938 | 0.931 | 0.931 | 0.933 | 0.932 | 0.900 | 0.900 |
| $D^{\mathrm{adv}}=25\%$ | $C=10, \gamma=1$ | **0.922** | 0.903 | 0.822 | 0.822 | 0.695 | 0.695 | 0.757 | 0.753 | 0.853 | 0.853 | 0.892 | 0.846 | 0.907 | 0.906 | 0.900 | 0.900 |

# I INTEGRATION WITH NEURAL NETWORKS

As demonstrated with the IMDB experiments in Section 4.1, FLORAL can be integrated with complex model architectures, e.g. a transformer-based language model such as RoBERTa, serving as a robust classifier head that enhances model robustness on classification tasks.

Similarly, FLORAL can be directly incorporated into neural networks by utilizing the last-layer embeddings (the $x_i$'s in Algorithm 1) as inputs. These extracted representations can then be trained using FLORAL, resulting in more robust feature representations. Notably, our theoretical analysis remains valid under this integration, ensuring the approach's soundness.

To demonstrate this further, we performed additional experiments on the Moon and MNIST-1vs7 (Deng, 2012) datasets, by integrating FLORAL with a neural network.

From Figure 17, we can conclude that FLORAL integration achieves a higher robust accuracy level compared to plain neural network training.

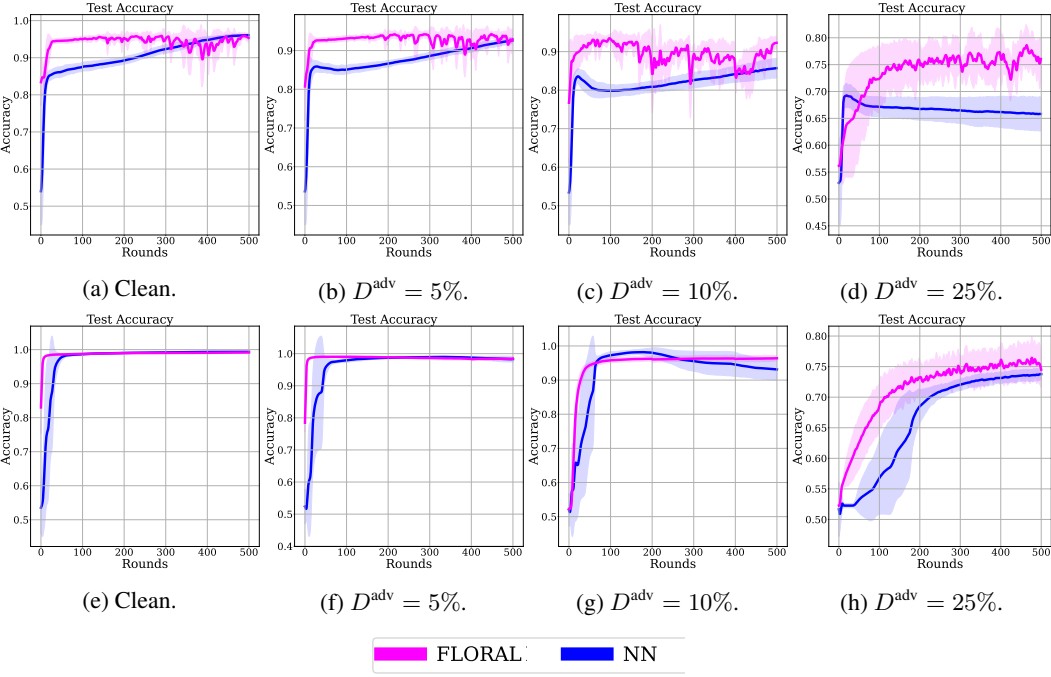

Figure 17: Clean and robust test accuracy performance of neural network vs FLORAL-integrated neural network trained on the Moon (**the first row**) and MNIST-1vs7 (**the second row**) datasets. The results demonstrate that FLORAL integration helps to achieve a higher robust accuracy level.

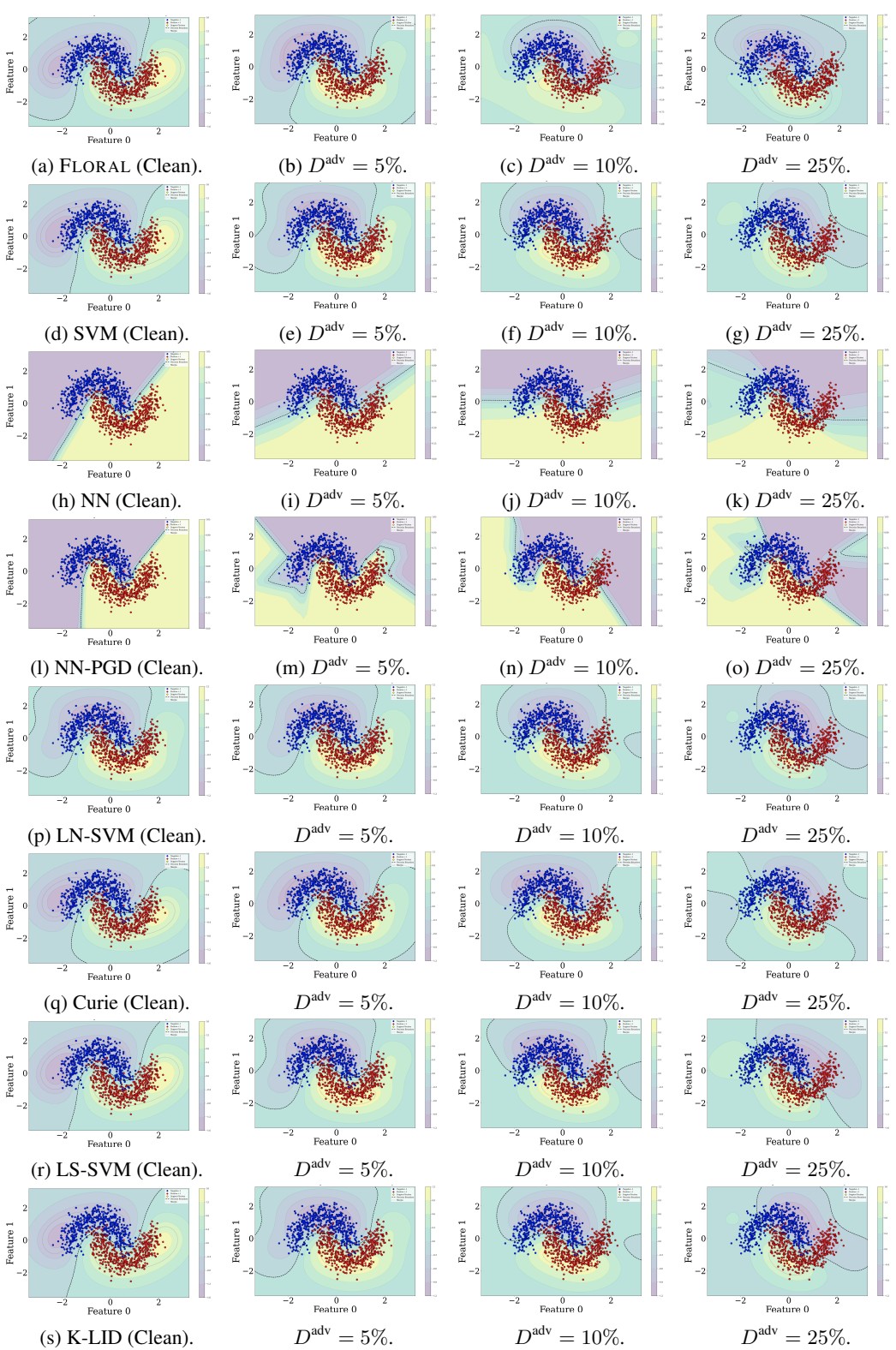

Figure 18: The decision boundaries on the Moon test dataset with various label poisoning levels. SVM-related models use an RBF kernel with $C = 10$ and $\gamma = 0.5$. FLORAL generates a relatively smooth decision boundary compared to baseline methods, particularly in $25\%$ adversarial setting, where baselines show drastic changes in their decision boundaries as a result of adversarial manipulations.

