# OpenReview forum: "Adversarial Training for Defense Against Label Poisoning Attacks"
_ICLR.cc/2025/Conference — ICLR 2025 Poster_

### Official Review · Reviewer_jdAy · 2024-11-01

**Soundness:** 2
**Presentation:** 2
**Contribution:** 2
**Rating:** 3
**Confidence:** 4

**Summary:**

This paper presents an adversarial training algorithm to mitigate label flipping attacks against Kernel SVM. Label flipping attacks aim to poison the training process of a Deep Neural Nets-based model via flipping the labels of training data. Similar as the vanilla adversarial training algorithm, the proposed method defines a bi-level optimization problem. The inner-layer problem is to choose the target training samples to flip their class label, in order to maximize the training loss of the classifier. The out-layer problem aims to minimize the worst-case classification loss facing the label flipping noise. This work presents a stability analysis to prove the convergence of the proposed method. Empirical study conducted over two datasets confirms the validity of the proposed method for enhancing the robustness of Kernelized SVM.

**Strengths:**

It is interesting to use the framework of adversarial training to defend label flipping attacks. The design of the proposed is further enhanced with a theoretical proof the convergence of the proposed method.

**Weaknesses:**

1. Label flipping attacks have been extensively investigated in the past years. This study lacks a comprehensive comparison to the recent advance in this field. I list a few as below [1,2,3].

[1]. Elan Rosenfeld, Ezra Winston, Pradeep Ravikumar, and J. Zico Kolter. 2020. Certified robustness to label-flipping attacks via randomized smoothing. In Proceedings of the 37th International Conference on Machine Learning (ICML'20)
[2]. Andrea Paudice, Luis Muñoz-González, and Emil C. Lupu. 2019. Label Sanitization Against Label Flipping Poisoning Attacks. In ECML PKDD 2018 Workshops.
[3]. P. Tavallali, V. Behzadan, A. Alizadeh, A. Ranganath and M. Singhal, "Adversarial Label-Poisoning Attacks and Defense for General Multi-Class Models Based on Synthetic Reduced Nearest Neighbor," 2022 IEEE International Conference on Image Processing (ICIP), Bordeaux, France, 2022, pp. 3717-3722.

2. Why Q is chosen as ${\tilde{y}_i}{\tilde{y}_j}K_{i,j}$ is not clearly justified.

3. It is unclear why using the dual variables $\lambda_i$ to choose training samples to flip their labels is better than using the influence function.

4. The proposed method seems limited only to Kernelized SVM, yet not generalizable to other model architectures.

**Questions:**

1. Include more defensive baselines into the comparative study, especially [1] mentioned in the weakness comment.

2. Clarify the bullet point 2 and 3 in the weakness comment.

3. Justify if the proposed method can be exntended to other model architectures. If possible, please demonstrate how the proposed method can be generalized with further empirical evaluations.

---

> ### Author Response · Authors · 2024-11-21
>
> We thank the reviewer for their comments. We are glad you found the FLORAL framework interesting and recognize its theoretical backup for its design.
> We address your concerns and questions below.
>
> ---
> - **[Baselines & Q1]** We first need to clarify that the reference [2] you provided, is already considered a baseline in our experiments. Please note that the baseline 4. LS-SVM (line 374, Section 4) is exactly the same method given in [2]. We cited their arXiv paper, however, we changed the citation to the one you provided in our updated manuscript.
> Prompted by your feedback, we further compared FLORAL against the methods [1,3], with results presented in Appendix J of the updated manuscript. As shown in Tables 11-14, the FLORAL approach significantly outperforms these baselines, with up to 40% improvement in accuracy level in most of the datasets. (e.g., in Table 14, approximately 45-50% improvement against RSRNN [3]; in Table 12 up to 17% improvement against RS method [1], etc.).
> ---
> - **[Q choice& Q2]** Q should be set as such because the attacker (inner problem) outputs a poisoned label set $\tilde{y}$, which the model (the outer problem) learns a soft margin SVM classifier **using the poisoned label set** $\tilde{y}$. Hence, compared to the classical formulation for a soft margin SVM classifier (formulation provided in Section 2 lines 117-121) (Boser et al., 1992; Hearst et al., 1998), the label set contains potentially poisoned labels. Because the features are not poisoned, the kernel matrix stays the same as $K_{ij}$ with the classical formulation for a soft margin SVM classifier.
> ---
> - **[Using dual variables & Q2]** Using SVM formulation and its dual variables **directly** provides us with an **interpretable** framework to **identify influential training points**. The dual variables explicitly quantify the contribution of each data point to the decision boundary. Influence functions often require approximations, by computing gradients or may require assumptions such as smoothness [4], which is eliminated when dual variables are used. In our paper, we only claim that using dual variables has the same intuition as using influence functions.
> ---
> - **[Generalizability & Q3]** As we already demonstrated in IMDB experiments with RoBERTa + FLORAL integration, FLORAL can be integrated with other model architectures, directly by using it as a robust classifier head. That is, extracting the last layer embeddings from the model e.g. RoBERTa or a neural network, etc. (which will be $x_i$’s in the algorithm), and then training with FLORAL. As shown in IMDB experiments, FLORAL integrated on top of complex model architecture such as RoBERTa achieves better robustness and supports **the adaptability of the FLORAL** framework in integration with other model architectures. We think that it is an interesting way to highlight FLORAL's effectiveness and usefulness in modern real-world classification scenarios, where FLORAL directly improves the robustness of some underlying complex model architectures (such as a pre-trained language model in the case of RoBERTa) on classification tasks. To demonstrate the integration further, we performed **additional experiments** on NN + FLORAL integration on the Moon and MNIST datasets. As given in Appendix-L in the updated manuscript, with FLORAL integration, NN achieves a higher robust accuracy level, e.g. see Figure 20-(d) that FLORAL provides approximately 10% improvement on the accuracy.
> ---
>
> Overall, the additional experiments have added depth to our empirical analysis, and further demonstrated FLORAL’s effectiveness against a diverse set of baselines and generalizability with other model architectures. We are happy to expand further if any parts remain unclear.
>
> ---
> > [1,2,3] are given in the reviewer’s comment.
>
>
> > [4] Koh, P. W., & Liang, P. (2017, July). Understanding black-box predictions via influence functions. In International conference on machine learning (pp. 1885-1894). PMLR.

---

> > ### Author Response · Authors · 2024-11-27
> >
> > Dear Reviewer jdAy,
> >
> > As we approach the end of the discussion period, we would like to ask if there is any additional information we can provide that might influence your evaluation of the paper. We believe our previous response with additional experiments conducted has effectively addressed your initial concerns and clarified some points that may have been overlooked.
> >
> > Thank you once again for your thoughtful consideration of our work.

---

### Official Review · Reviewer_t4Fm · 2024-11-03

**Soundness:** 3
**Presentation:** 2
**Contribution:** 2
**Rating:** 6
**Confidence:** 3

**Summary:**

The paper introduces FLORAL, a defense mechanism against label-flipping poisoning attacks targeting machine learning (ML) models. The authors argue that these poisoning attacks, which alter the training data labels to mislead model learning, pose a more challenging defense problem than attacks based on feature manipulation.

The FLORAL defense is based on the formulation of the training problem as a non-zero-sum Stackelberg game. This novel approach transforms the learning problem under poisoning attacks into a bilevel optimization problem. The training is performed during several rounds, during which an attacker tries to maximize the damage induced by label flipping while the training algorithm minimizes the loss of the dataset poisoned by the attacker. FLORAL uses a projected gradient descent algorithm to solve this optimization problem, tailoring the solution specifically to support vector machines (SVMs) while indicating that it can be extended to neural networks. The authors provide theoretical proof that model parameter updates by FLORAL remain bounded and that the algorithm converges to the Stackelberg equilibrium.

Regarding experimental validation, FLORAL is tested on both a synthetic dataset and the IMDB sentiment analysis dataset, using seven established defenses as baselines. On the synthetic dataset, a SVM protected by FLORAL significantly outperforms these baseline defenses, maintaining robustness even with 40% of poisoned labels. For the IMDB dataset, FLORAL-defended SVM is compared against a RoBERTa language model. The results demonstrate that an SVM trained with FLORAL on the poisoned IMDB subset performs more robustly than a poisoned RoBERTa model as the percentage of poisoned labels increases.

**Strengths:**

Thanks to the authors for submitting this interesting work. The proposal of the paper is for sure original and novel since the problem of defending an ML model against a label flip poisoning attack is cast as an adversarial training problem for the first time. The specific formulation of the problem as a non-zero sum Stackelberg is also original and enables the design of a defense algorithm that outperforms existing defenses. Moreover, the theoretical analysis behind the convergence to the optimum of the proposed training algorithm is good. Finally, regarding the experimental evaluation, it is appreciable that the experimental evaluation considers several state-of-the-art defenses. In this way, it is possible to assess that the proposed defenses overcome several defenses and not some specific ones.

**Weaknesses:**

A key weakness of the paper lies in the lack of clarity about the motivation for using adversarial training to counter poisoning attacks and for formulating the problem as a non-zero-sum Stackelberg game. Adversarial training, particularly in the context of evasion attacks on ML models, typically aims to train the model on possible attacks to make it robust against test time attacks, as explained in Sections 1 and 3.3 in this paper. Instead, defenses against poisoning attacks often focus on removing or mitigating the impact of poisoned samples once the attack _has already been performed_. Given this distinction, the paper does not sufficiently explain why including a simulated poisoning attack within the training procedure should diminish the impact of already poisoned samples. The motivation seems clear when the training set is clean: the training procedure takes into account the worst attacks that the attacker can perform using the clean training set. However, in practice, the defense algorithm will most often be applied to an already poisoned dataset. In this scenario, the attacker in the game flips labels that can have _already_ been flipped, disrupting the initial attack performed to poison the dataset. If it is the reason behind the efficacy of the defense, the authors should state it clearly in the paper. Otherwise, they should provide an intuition behind the efficacy of the attack when it is applied to a poisoned training set. Moreover, it is not clear the meaning of the equilibrium of the game in the context of defending against poisoning attacks. The authors should also clarify this aspect.

The experimental evaluation also appears to have several notable shortcomings that should be addressed to strengthen the validation of the proposed defense. First, the attacks used to generate poisoned datasets do not represent the state-of-the-art. While the authors demonstrate that their defense outperforms existing defenses against the relatively simple attack on the synthetic dataset, FLORAL is less effective than the baselines against the Alfa attack (Xiao et al., 2015) when considering more realistic attack budgets (e.g., 5% and 10%). While it is reasonable that a single defense may not counter every attack type, the authors should examine if the defense is also less effective than other considered defenses when considering other attacks proposed in literature, like the ones in (Biggio et. al., 2011) and (Paudice et. al., 2018). Furthermore, the effectiveness of the defense against the baselines is shown only on two dataset. To demonstrate that the defense generalizes well, the authors should consider testing it on additional datasets, as done in (Biggio et al., 2011) and (Paudice et al., 2018). Finally, the comparison between the robustness of FLORAL-defended SVMs and RoBERTa on the IMDB dataset is not fair, in my opinion. The authors states that `Adversarial datasets were created by fine-tuning the RoBERTa-base model on the clean dataset to identify influential training points` (Section 5). This approach inherently biases the attack towards RoBERTa, as it is crafted specifically to target that model. Therefore, the finding that an SVM defended by FLORAL outperforms RoBERTa in terms of robustness on a poisoned dataset may not be reliable, as the attack is tailored to RoBERTa. I suggest that the authors perform the comparison in a fairer way, for example by creating adversarial datasets by flipping the most influencial points for an SVM trained on IMDB.

Finally, the presentation of the paper is almost good, but there are a few unclear, unsupported, or potentially misleading statements that need clarification:
- In the caption of Figure 2, the authors state that `label poisoning affects any point in the input space uniformly`. However, this statement seems counterintuitive. Intuitively, flipping the labels of points closer to the decision boundary should have a greater impact on shifting the boundary than flipping labels of points far from it.
- In Section 3, the authors claim that `In the case of an SVM-based classifier, the minimax formulation would only safeguard against
attacks that target data points very far from the decision boundary. These attacks are unlikely to alter the SVM classifier significantly, as such points are less likely to be support vectors, i.e. the critical data points that define the decision boundary.`. However, no references or further explanation are provided to substantiate this observation.
- The authors describe the considered attacker in the game as white-box. However, they subsequently state that `However, in practice,
the attacker’s knowledge of the model may be limited. To account for this, we assume the attacker operates under a constrained budget, allowing the poisoning of at most B labels per round` (Section 3). This constraint does not limit the attacker’s knowledge (which remains white-box) but rather restricts the attacker’s capability to flip labels within a budget.

**Update after authors' response**
The authors did a good job in responding to the questions below and clarifying the weaknesses described above. I will leave the weaknesses written to keep a trace, but the authors clarified them. In particular, the authors clarified the formalization of the problem as a non-zero-sum Stackelberg game, included other attacks in their evaluation with positive results, and clarified the experimental setting of the experiments with RoBERTa, as suggested. Moreover, they clarified other statements listed above.

**Arguments for the score**

The formulation of the problem of defending against label flip attacks as a non-zero-sum Stackelberg game is novel, and the experimental results support the effectiveness of the proposed defense. However, there are still some limitations in the proposed approach, which are also highlighted by other reviewers. In particular, it is not clear if the proposed adversarial training method can be generalized to more popular model architectures, e.g., Neural Networks. Indeed, you have shown that you can attach your adversarially trained SVM as a head of a more complex model, but not if your adversarial training method generalizes. Finally, the proposed method does not always allow the model to achieve better accuracy under attack with respect to other baselines when the budget is realistic (5-10% of poisoned data), even though these cases are sporadic.

**Questions:**

- What are the motivations for employing an adversarial training procedure and formulating the defense problem as a non-zero-sum Stackelberg game? (See the Weaknesses section for further details.)
- What precisely does the Stackelberg equilibrium represent within the FLORAL algorithm?
- Could the algorithm described in Section 3.2 return a projection that does not satisfy Constraint 6 of the problem formulation? If so, what implications would this have for the convergence of the training algorithm?
- Can the authors address the concerns regarding the experimental evaluation? (See the Weaknesses section for further details.)
- Could the authors clarify the unclear and unsupported statements listed in the Weaknesses section?

**Details Of Ethics Concerns:**

.

---

> ### Author Response · Authors · 2024-11-21
>
> Thank you for your thoughtful feedback. We are happy that you recognize the originality and novelty of our proposed framework, as well as appreciate the theoretical analysis and presentation provided in our paper. We address your concerns and questions below, including additional experiments and paper edits inspired by your comments.
>
> ---
> - **[AT defense & Q1]** As noticed by the reviewer, when the training set is clean, adversarial training (AT) takes into account the worst attacks created by the attacker using the clean training set. Using these adversarial examples, the model is trained to avoid overfitting and to reduce the sensitivity to a small set of critical training points, which improves robustness. Thus, the FLORAL approach directly reduces the sensitivity of the model output on single training points, which, as can be argued from the perspective of algorithmic stability, helps generalization. When the training set is already poisoned, FLORAL is effective in identifying already poisoned labels. Indeed the reviewer’s intuition about FLORAL disrupting the initial attack is correct. To further demonstrate this with a quantitative analysis, we provided the percentage of recovered poisoned labels by FLORAL in Figure 19, and a discussion in Appendix K of the updated manuscript. As shown, FLORAL, on average, disrupts 30% of the initial attack, while only (erroneously) affecting a small number of unpoisoned data points. We evaluated this over different hyperparameter choices and showed that FLORAL consistently achieves this rate. We also demonstrate an example trace of a round in Figure 19, illustrating which points are initially poisoned and which of them are recovered by FLORAL on the Moon dataset.
> - - Also, note that we have already shown a similar quantitative analysis with IMDB experiments, see Figure 10-(a) in Appendix D.2, that FLORAL maintains (on average) >50% of the influential training points. When the training set gets more adversarial, the recovery rate decreases naturally. However, even in the most adversarial setting with 40% adversarial labels, FLORAL maintains >50% of the influential training points for the model’s predictions. Compared to Figure 10-(b), the RoBERTa model not integrated with FLORAL shows a significant change in the influential training points. This clearly demonstrates that FLORAL diminishes the impact of already poisoned samples.
> ---
> - **[Non-zero-sum Stackelberg game formulation]** Our formulation relies on the following key points (which we explain in Section-3):
> - - As shown by Robey et al. (2024), the min-max formulation for AT does not capture the attacker’s objective correctly, since from the optimization perspective, the attacker should maximize a lower bound of the loss function. This motivates the non-zero-sum formulation.
> - - Our formulation is a Stackelberg game since the attacker generates the label attack using the dual variables of the model. The interplay starts with initial model parameters $\lambda$, and then the attacker takes action $(\tilde{y}(\lambda))$ based on observed model parameters $\lambda$. This creates a leader-follower type of game (see Figure 3 which provides a graphical summary of this interplay).
> ---
> - **[Equilibrium & Q2]** The equilibrium stands for the best-case outcome for the defender when facing the worst-case poisoning attack. At equilibrium, the attacker cannot find a better label attack without worsening the defender’s response, and vice versa.
> ---
> - **[Additional attacks & Q4]** We further evaluated FLORAL defense against alfa-tilt [1] and LFA attack [2] from the literature and report the results in Appendix H-I. These analyses further support FLORAL’s effectiveness against diverse attacks from the literature.
> - - Some of the baselines (e.g. K-LID (Weerasinghe et al., 2021), LN-SVM (Biggio et al., 2011) )) indeed consider attack rates 30%-40%. FLORAL consistently outperforms or is on par with all baselines, except for the alfa attack for smaller attack rates. For larger attack rates FLORAL significantly outperforms all baselines (including alfa), while many of the baselines fail.
> ---
> - **[Additional dataset & Q4]** We provide extensive additional experiments on the MNIST dataset in Appendix-G. We also experimented on MNIST using the alfa-tilt attack (see Appendix-I). These analyses showed that FLORAL achieves higher robust accuracy compared to baseline methods (typically improving by e.g. 4.5% against the next best method in the most adversarial setting (see Figure 18-d)). We now clearly show FLORAL **generalizes well** across a diverse set of datasets from numerical, text, and vision contexts, under different label poisoning attacks.

---

> ### Author Response · Authors · 2024-11-21
> **rebuttal-continue**
>
> - **[Comparison with RoBERTa]** The experiments with RoBERTa show the **adaptability** of the FLORAL framework in **integration with other model architectures**, and how it improves the robustness of those models against the label poisoning attacks carefully crafted for such models. This highlights FLORAL's effectiveness and usefulness in modern real-world classification scenarios, where FLORAL directly improves the robustness of some underlying complex model architectures (such as a pre-trained language model in the case of RoBERTa) on classification tasks. We do not agree with the reviewer's comment that attacks should not be targeted toward the RoBERTa model. Because safeguarding against targeted attacks is indeed highly relevant for the application scenario we have in mind. Based on your comment, we further clarified the aim of this experiment in the text and updated our manuscript accordingly (lines 463-464).
> ---
> - **[Label poisoning statement & Q5]** When we consider **classification accuracy** as the measure, then given (fixed) a classifier $f(x) \to y$, flipping the label of any data point, i.e. $y$ to $\tilde{y}$, would contribute to the accuracy by -1. (As x is the same, our fixed classifier assigns the label y). Whereas, in a feature attack case: $x \to \tilde{x}$, the classifier $f$ can assign a different label depending on the perturbed $\tilde{x}$. Hence, the effect on the accuracy is not uniform.
> ---
> - **[Statement for the minimax formulation & Q5]** In the case of a minimax formulation with Hinge loss, the attacker creates the worst possible attack by focusing on the points farthest from the decision boundary on the wrong side, as these would incur the highest loss. However, in a soft-margin SVM, the decision boundary is determined by the support vectors, which are typically closer to the boundary. These farthest-point attacks are unlikely to have a substantial impact on the boundary itself as soft-margin SVMs are shown to be robust to outliers (see [4] Section 1.5)
> ---
> - **[White-box vs capability statement & Q5]** Having read the statement with your comment, we agree with the reviewer. We have updated the statement as “the attacker’s knowledge of the training data…”, since adding a budget implicitly simulates the case when the attacker cannot alter all of the training labels.
> ---
> - **[Q3]:** In Algorithm 2, the resulting $\mu$ is an approximation to the optimal one (for large-scale settings), hence, the constraint (6) may be satisfied with some error. This reflects the global convergence as an additional $\frac{\epsilon}{1-\kappa_{\lambda}}$ term to the right-hand side of the Equation (33) of the updated manuscript, where $\epsilon$ is the error amount. Thus, the global convergence guarantee still applies for approximate $\mu$ (provided that $\delta'$ is adjusted accordingly and that the approximation error in $\mu$ is bounded).
> ---
>
> Overall, we thank you for the constructive feedback. With the additional analysis we provided, we extensively analyze FLORAL’s performance across a diverse set of baselines, against a diverse set of label attacks and on different types of datasets.
>
> We are happy to expand on the above points further if further questions remain.
>
> ---
> > [1] Xiao, H., Biggio, B., Nelson, B., Xiao, H., Eckert, C., & Roli, F. (2015). Support vector machines under adversarial label contamination. Neurocomputing, 160, 53-62.
>
> > [2] Andrea Paudice, Luis Muñoz-González, and Emil C. Lupu. (2019). Label Sanitization Against Label Flipping Poisoning Attacks. In ECML PKDD 2018 Workshops.
>
> > [3] Robey, A., Latorre, F., Pappas, G. J., Hassani, H., & Cevher, V. (2024). Adversarial training should be cast as a non-zero-sum game. International Conference on Learning Representations (ICLR'24)
>
> > [4] Schölkopf, B. (2002). Learning with kernels: support vector machines, regularization, optimization, and beyond.

---

> > ### Comment · Reviewer_t4Fm · 2024-11-24
> > **Thank you for your response; I have additional questions**
> >
> > Thanks a lot to the authors for addressing the weaknesses and questions raised in my review. The additional experimental results have been helpful in highlighting the good performance of FLORAL concerning other well-known baselines. Moreover, the answers to the questions about the formulation of the problem make the formulation more clear now. I have these additional questions and observations about your response:
> > -  **[Label poisoning statement & Q5]**:  I respectfully do not agree with the response to this point. Changing the label of a training point may not impact the accuracy of -1. Indeed, changing the label of an (attacked) training point may also induce no misclassification, i.e., the corresponding training point in the clean training set still continues to be correctly classified. Furthermore, it may induce more misclassifications than one, depending on the training algorithm. Let me know if I have misunderstood something about your response.
> > - **[White-box vs capability statement & Q5]**: Thanks for trying to correct the statement in the paper. However, I do not agree with your correction. If the attacker uses a specific budget, this does not imply that it has limited knowledge of the training data in the context of your attack. As far as I understood, the attack flips at most B labels per round, and the labels are selected over the _entire_ training set. Then, if the attacker had limited knowledge about the training set, it would be able to select the labels to attack always from the same _fixed subset_ of the training set over _all the steps_ of the attack. The constraint on the budget instead limits the attacker's capabilities, e.g., it changes a few labels to remain stealthy.
> > - **[AT defense & Q1]**: What does `percentage of recovered poisoned labels` precisely mean? Do you mean that the model correctly classifies the training points in the clean training set that were poisoned by the attack? I ask this only to be sure about the interpretation.

---

> ### Author Response · Authors · 2024-11-25
> **Further clarification**
>
> We further address your questions below.
> - **[Label poisoning statement & Q5]:** Under a perfect classifier (tries to achieve 100% classification accuracy), the effect of a label attack on the classification accuracy would be **+1** or **-1**, depending on whether the point is already poisoned or not, however, still irrelevant where the point is located (uniform over the location relative to the decision boundary).
> - **[White-box vs capability statement & Q5]:**  We safeguard against the worst-case attacker and assume that this attacker has access to the labels of the B support vectors, which creates limited access to training points. In each iteration of FLORAL, the attacker has access to B support vectors, from which it can poison $k$ (with limited capability). With our statement, we aimed to only “mimic” the scenario in practice.
> - **[AT defense & Q1]:** The percentage of recovered poisoned labels quantifies how many of the poisoned training points are flipped back to their original label by the FLORAL framework. For instance, in the situation where 5% of the data points are poisoned, FLORAL is consistently reversing the labels of 30% of these, which means that at each iteration FLORAL is much more likely to revert the labels of poisoned data points than to erroneously flip the labels of unpoisoned ones.
>
> We hope that these points clarified your questions.

---

> ### Author Response · Authors · 2024-11-26
>
> Dear Reviewer t4Fm,
>
> As we approach the end of the discussion period, we would like to ask if there is any additional information we can provide that might influence your evaluation of the paper. We believe our previous response with additional experiments conducted & paper edits prompted by your feedback, has effectively addressed your initial concerns and clarified some points that may have been overlooked.
>
> Thank you once again for your thoughtful consideration of our work.

---

> > ### Comment · Reviewer_t4Fm · 2024-11-26
> > **Thank you for your clarifications**
> >
> > Thanks a lot to the authors for providing additional clarifications. I will update my score and continue the discussion with the other reviewers.

---

> > > ### Author Response · Authors · 2024-11-27
> > > **Thanks!**
> > >
> > > Thank you so much for raising your score and supporting our paper! We are happy to hear that you find our additions and clarifications helpful for the readers. We sincerely appreciate your thoughtful feedback which has helped us improve our paper.

---

### Official Review · Reviewer_bjpg · 2024-11-04

**Soundness:** 2
**Presentation:** 3
**Contribution:** 2
**Rating:** 3
**Confidence:** 3

**Summary:**

This paper presents FLORAL, an adversarial training defense strategy that utilizes support vector machines (SVMs) to mitigate label poisoning attacks. FLORAL frames the training process as a non-zero-sum Stackelberg game, positioning an attacker who strategically poisons key training labels against the model’s objective of resilience. The FLORAL employs a projected gradient descent algorithm with kernel SVMs to improve the robustness of adversarial training.

**Strengths:**

1. The paper is well-presented, with clear and high-quality writing.
2. The figures are visually appealing.
3. The FLORAL achieves  SOTA  performance compared to the selected baselines.

**Weaknesses:**

1. All baselines examined in this paper were published before 2021. To support the claim of achieving state-of-the-art performance, the authors may consider including comparisons with more recent methods.
2. Have the authors considered placing the figures and tables at the top of the current page?
3. The authors mention using ``SVM with an RBF kernel, which serves as a basic benchmark (Hearst et al., 1998).'' Have more recent approaches been considered as benchmarks?
4. Could the authors add a comparison with adversarial training methods?

**Questions:**

1. Please use the reference citation format specified in ICLR.
2. The datasets in this paper are all binary classification datasets. Does this approach apply exclusively to binary classification tasks, or could the authors provide additional results on non-binary classification datasets to expand the analysis?
3. The Moon and IMDB datasets represent entirely different types of data. Could the authors clarify the preprocessing steps applied to each dataset in the experiments? Currently, it is unclear how the proposed method effectively applies to such diverse data types？
4. The results in Table 1 show that the 1998 method, SVM, is the second-best performing method, following only the proposed approach. This is somewhat surprising, as more recent methods from 2021 or 2018 would be expected to outperform an older approach, especially given their likely state-of-the-art status upon publication. Could the authors provide an explanation for this finding?
5. RoBERTa is widely regarded as a pre-trained language model in the NLP community. Could the authors explain the rationale for using RoBERTa as a baseline in this study?

---

> ### Author Response · Authors · 2024-11-21
>
> We thank the reviewer for the comments. We are glad you appreciate the presentation, writing, and performance of our approach. We address the raised concerns and questions below:
>
> ---
> 1. **[Baselines]** The most recent baselines addressing defense against label poisoning indeed date back to 2021-2022, a fact also pointed out by **reviewer jdAy** (see the corresponding references). We further compared our approach against recent methods [1] and [2], with results presented in Appendix J of the updated manuscript. As shown in Tables 11-14, the FLORAL approach significantly outperforms these baselines, with up to 40% improvement in accuracy level in most of the datasets.
> 2. We changed the figure and table placement, particularly for pages 9-10.
> 3. We compared FLORAL against 8 diverse baselines + 2 additional provided [1,2], ranging from fundamental approaches to state-of-the-art defense methods.
> 4. As stated in our contributions in Section 1, to the best of our knowledge, this is the first work to introduce adversarial training (AT) as a defense for label poisoning attacks. Hence, there is no further AT method that **specifically addresses defending against label poisoning attacks.** We already included and compared our method against an AT method (NN-PGD) which defends against feature perturbation attacks, we think adding extensive comparison against other AT defense feature perturbation methods is beyond the scope and not aligned with the main problem we address in our paper, which is a defense against label poisoning attacks.
> ---
> - **Q1:** We used the reference citation format (“\bibliographystyle{iclr2025_conference}”) in our paper. Presumably, the coloring was the concern (which we would find somewhat surprising though?), we changed the coloring to the usual color in the updated manuscript and hope this clarifies the issue.
> - **Q2:** Our approach is indeed not restricted to binary classification tasks and can be extended to multi-class classification settings using a one-vs-all kernelized SVM strategy. To ensure clarity, we have already provided Algorithm-3 in Appendix-E. Given the straightforward nature of this extension, the details we presented provide a sufficient foundation for applying our approach to multi-class classification.
> - **Q3:** The Moon dataset is a 2-dimensional synthetically generated dataset (illustrated in Figure 7 in Appendix C.1). We created train/val/test splits for 5 replications and for each setting, we generated adversarially labeled dataset by flipping the labels of points farther from the decision boundary of a linear classifier trained on the clean dataset (see Section-4). For the IMDB dataset, we extract embeddings (768-dimensional numerical representations) for the data points using the RoBERTa language model. We first fine-tune RoBERTa model on the IMDB dataset. Then, we extract the last layer embeddings of the fine-tuned model which serve as embeddings for the data points in the IMDB dataset.
> - **Q4:** Soft-margin SVMs are shown to be robust to outliers (see [3] Section 1.5), however, as the dataset gets more adversarial (see 25% setting), the influence of individual patterns (adversarial examples) gets more severe. Hence, the reviewer’s observation no longer consistently holds. Also, as given in additional results in Appendix D.1 and Appendix G, Vanilla SVM does not consistently perform well in all settings.
> - **Q5:** The experiments with RoBERTa show the **flexibility and adaptability of the FLORAL** framework in integration with other model architectures, and how it improves the robustness of those models against label poisoning attacks. This highlights FLORAL's effectiveness and usefulness in modern real-world classification scenarios, where FLORAL directly improves the robustness of some underlying complex model architectures (such as a pre-trained language model in the case of RoBERTa) on classification tasks.
> ---
> Overall, with the additional analysis we provided, we extensively analyze FLORAL’s performance across a diverse set of baselines, against a diverse set of label attacks and on different types of datasets. The results highlight consistent and significant performance improvements against state-of-the-art baselines (e.g., in Table 14, approximately 45-50% improvement against RSRNN [2]; in Table 12 up to 17% improvement against RS method [1], etc.).
>
> We are happy to expand on the above points further if further questions remain.

---

> > ### Author Response · Authors · 2024-11-21
> > **references**
> >
> > > [1] Elan Rosenfeld, Ezra Winston, Pradeep Ravikumar, and J. Zico Kolter. 2020. Certified robustness to label-flipping attacks via randomized smoothing. In Proceedings of the 37th International Conference on Machine Learning (ICML'20)
> >
> >
> > > [2] P. Tavallali, V. Behzadan, A. Alizadeh, A. Ranganath and M. Singhal, "Adversarial Label-Poisoning Attacks and Defense for General Multi-Class Models Based on Synthetic Reduced Nearest Neighbor," 2022 IEEE International Conference on Image Processing (ICIP), Bordeaux, France, 2022, pp. 3717-3722.
> >
> > > [3] Schölkopf, B. (2002). Learning with kernels: support vector machines, regularization, optimization, and beyond.

---

> ### Author Response · Authors · 2024-11-27
>
> Dear Reviewer bjpg,
>
> As we approach the end of the discussion period, we would like to ask if there is any additional information we can provide that might influence your evaluation of the paper. We believe our previous response with additional experiments conducted has effectively addressed your initial concerns and clarified some points that may have been overlooked.
>
> Thank you once again for your thoughtful consideration of our work.

---

### Official Review · Reviewer_1cTC · 2024-11-05

**Soundness:** 2
**Presentation:** 3
**Contribution:** 2
**Rating:** 6
**Confidence:** 2

**Summary:**

This paper aims to solve the label poisoning attacks by formulating an adversarial training defense strategy, FLORAL based on SVM. The adversarial training is cast to a non-zero-sum Stackelberg game with an attacker. The projected gradient descent (PGD) is proposed to solve this problem. The experiments on two datasets show the robustness of the proposed method.

**Strengths:**

- The notation in this paper is generally clear, making the formula easy to follow.
- The plot illustration is clear.
- The theoretical proof and analysis is clear and promising.

**Weaknesses:**

- The experiments are limited. The datasets are relatively small.
- The author claims that FLORAL can be integrated with neural network but I did not find it. The proof here is all based on SVM and I suspect the proof for neural network, especially for the non-convex neural network, will be different. If the framework can only work and guarantee on SVM-based classifier, the contribution will be decreased a lot.
- Since it is a general method, the vision dataset, even MNIST, should be given.
- One of the contribution is projected gradient descent. The discussion about why it works should be given.

**Questions:**

- What is meaning of $\mathcal{I}_C$ and $\mathcal{I}_z$ in Line 294?
- More details about Moon dataset should be given. I especially interested in the multiple classification tasks and I hope more details about the experiment details can be given.

---

> ### Author Response · Authors · 2024-11-21
>
> We thank the reviewer for their feedback. We are glad you found our presentation and theoretical analysis clear and promising. We address your concerns below:
>
> ---
> - **[Additional experiments]** Prompted by your feedback, we have performed additional experiments on the MNIST dataset and reported the results in Appendix-G in the updated manuscript. The results on MNIST are consistent with the other datasets, further showing that FLORAL maintains a higher robust accuracy level when the dataset gets more adversarial. We now show FLORAL’s effectiveness evaluated on diverse datasets from numerical, text, and vision contexts. Furthermore, we considered additional label attacks from literature LFA [1] and alfa-tilt [2] on the Moon and MNIST datasets and evaluated the performance of FLORAL defense against all baselines, as given in Appendix-H-I. Doing so, we now show FLORAL’s performance with extensive experiments on a diverse set of attacks: randomized top-k support vector attack, alfa, lfa, alfa-tilt, and a gradient-based attack. Regarding the size of the datasets, IMDB has 50000 data points including train, validation, and test datasets. IMDB  and MNIST are the benchmark datasets used to evaluate the defenses against label poisoning attacks, as also given in the example references provided by the reviewers jdAy and t4Fm. These additional experiments further highlight that FLORAL consistently outperforms the state-of-the-art. More precisely, e.g. in Figure 18, FLORAL defends the best compared to baselines, which also shows its generalizability under different label attack types.
> ---
> - **[Integration with NNs]** The direct integration of FLORAL with neural networks, is to use FLORAL as a robust classifier head, as we have shown in the IMDB experiments with RoBERTa + FLORAL integration. That is, extracting the last layer embeddings from an NN model (which will be $x_i$’s in the algorithm), and then training with FLORAL. This way FLORAL helps to learn robust representations. We have already demonstrated in IMDB experiments that FLORAL integrated on top of complex model architecture such as RoBERTa achieves better robustness. The theoretical analysis still holds in this way of integration, as we have no assumptions regarding the inputs. To demonstrate the integration further, we performed **additional experiments** on NN + FLORAL integration on the Moon and MNIST datasets. As given in Appendix-L in the updated manuscript, with FLORAL integration, NN achieves a higher robust accuracy level, e.g. see Figure 20-(d) that FLORAL provides approximately 10% improvement on the accuracy.
> ---
> - **[Additional dataset]** Extensive additional experiments on MNIST presented in Appendix-G, I and H in the updated manuscript further showed that FLORAL achieves higher robust accuracy compared to baseline methods (typically improving by e.g. 4.5% against the next best method in the most adversarial setting (see Figure 18-d)).
> ---
> - **[PGD]** We have provided a thorough theoretical analysis of the projected gradient descent algorithm, including its convergence to the global optimum of the proposed training algorithm in Appendix A1-A4.
> ---
> - **Q1:** $\mathcal{I}_C$ is the index set for the data points having dual variable value $\geq C$. And, $\mathcal{I}_z$ is the index set for the data points having dual variable value $\in (0,C)$. We apply variable splitting and create these index sets to compute $\mu$ values, which then helps us to quantify the optimal $\lambda^{\star}$ value. We explain this further in the theoretical analysis given in Appendix-A1 (lines 876-877).
> ---
> - **Q2:** The Moon dataset is a 2-dimensional synthetic numerical dataset in which each feature takes value in the range $[-2.5,2.5]$. We provide the details (in Section 4) and its illustration in Appendix-C.1.
> ---
> - - **Q2.2:** We are unsure about what the reviewer means by “multiple classification tasks”. If multi-class classification is meant, then, our approach is indeed not restricted to binary classification tasks and can be extended to multi-class classification settings using a one-vs-all kernelized SVM strategy. To ensure clarity, we have already provided Algorithm-3 in Appendix-E. Given the straightforward nature of this extension, the details we presented provide a sufficient foundation for applying our approach to multi-class classification.
> ---
>
> Overall, with the additional analysis we provided, we extensively analyze FLORAL’s performance across a diverse set of baselines, against a diverse set of label attacks and on different types of datasets. We are happy to expand on the above points further if further questions remain.
>
> ---
> > [1] Andrea Paudice, Luis Muñoz-González, and Emil C. Lupu. (2019). Label Sanitization Against Label Flipping Poisoning Attacks. In ECML PKDD 2018 Workshop.
>
> > [2] Xiao, H., Biggio, B., Nelson, B., Xiao, H., Eckert, C., & Roli, F. (2015). Support vector machines under adversarial label contamination. Neurocomputing, 160, 53-62.

---

> ### Author Response · Authors · 2024-11-26
>
> Dear Reviewer 1cTC,
>
> As we approach the end of the discussion period, we would like to ask if there is any additional information we can provide that might influence your evaluation of the paper. We believe our previous response with additional experiments conducted has effectively addressed your initial concerns and clarified some points that may have been overlooked.
>
> Thank you once again for your thoughtful consideration of our work.

---

> ### Author Response · Authors · 2024-11-27
>
> Thank you so much for raising your score and the kind words. We are happy to hear that the reviewer appreciates our clarifications. Your feedback has helped us improve our paper, and we sincerely appreciate that.

---

### Author Response · Authors · 2024-11-21
**General Response**

We thank the reviewers for their time, positive comments, and constructive feedback.

The reviewers found our approach to be original and novel (`“The proposal of the paper is for sure original and novel”` (t4Fm) and `"It is interesting to use the framework of adversarial training to defend label flipping attacks"` (jdAy)), with a clear theoretical analysis (`“The theoretical proof and analysis is clear and promising.”`(1cTC, jdAy)), and it was regarded to be well-written (`“The paper is well-presented, with clear and high-quality writing.”`(bjpg)), and that the proposal is experimentally supported against diverse SOTA defenses (`“it is appreciable that the experimental evaluation considers several state-of-the-art defenses”`(t4Fm)).

In summary, the main contribution of our work is to introduce FLORAL: a novel adversarial training defense mechanism against label poisoning attacks targeting machine learning models.

---
Considering the reviewer’s comments, we updated our manuscript (modified parts **highlighted in “red”**), with the following additions:

1. **[Additional baselines]** We provide further comparisons against additional baselines [1,2] in Appendix J (see Tables 11-14) of the updated manuscript. Our method is shown to outperform these baselines significantly on several datasets, with up to 40% improvement in accuracy level.
2. **[Additional label poisoning attacks]** We show the generalizability of the FLORAL approach against additional label poisoning attacks: LFA, alfa-tilt from the literature [3,4]. We report the results in Appendix H-I. The results further substantiate the performance benefits of FLORAL. Particularly, the results on MNIST under the alfa-tilt attack support that FLORAL (even though it is not designed against this attack) is most effective against baselines and it generalizes well.
3. **[Additional dataset]** We further evaluate FLORAL against baseline methods on the MNIST dataset, for which we provide the performance comparison in Appendix-G. The results on MNIST are consistent with the other datasets, further showing that FLORAL maintains a higher robust accuracy level when the dataset gets more adversarial.
4. **[Integration with other model architectures]** We discuss and provide additional analysis on FLORAL integration with other model architectures in Appendix-L, and also in Section 4.1 (lines 463-467). With FLORAL integration, neural network and RoBERTa models achieve a higher robust accuracy level, e.g. see Figure 20-(d) that FLORAL provides approximately 10% improvement in the accuracy.

Once again, we sincerely thank all reviewers for their thoughtful and constructive comments, which have helped us improve the quality of our work.

---
> [1] Elan Rosenfeld, Ezra Winston, Pradeep Ravikumar, and J. Zico Kolter. 2020. Certified robustness to label-flipping attacks via randomized smoothing. In Proceedings of the 37th International Conference on Machine Learning (ICML'20)

> [2] P. Tavallali, V. Behzadan, A. Alizadeh, A. Ranganath and M. Singhal, "Adversarial Label-Poisoning Attacks and Defense for General Multi-Class Models Based on Synthetic Reduced Nearest Neighbor," 2022 IEEE International Conference on Image Processing (ICIP), Bordeaux, France, 2022, pp. 3717-3722.

> [3] Andrea Paudice, Luis Muñoz-González, and Emil C. Lupu. 2019. Label Sanitization Against Label Flipping Poisoning Attacks. In ECML PKDD 2018 Workshops.

> [4] Huang Xiao, Battista Biggio, Blaine Nelson, Han Xiao, Claudia Eckert, and Fabio Roli. Support vector machines under adversarial label contamination. Neurocomputing, 160:53–62, 2015.

---

### Meta-Review · Area_Chair_EY2i · 2024-12-25

**Metareview:**

This paper proposes  FLORAL, an adversarial training defense strategy to label poisoning attacks based on SVM.
The reviewers acknowledge that the paper is well-written and the figures are clear and visually appealing.
The formulation of the problem and the solution are novel, and the experimental results show the superior performance of FLORAL. The authors addressed most of the concerns raised by the reviewers, although some of the reviewers did not respond and update the scores accordingly. Given the above reasons, I recommend accept.

**Additional Comments On Reviewer Discussion:**

The authors addressed most of the concerns raised by the reviewers, although some of the reviewers did not respond and update the scores accordingly.

---

### Decision · Program_Chairs · 2025-01-22

Accept (Poster)